# SUPERVISED AND SEMI-SUPERVISED DIFFUSION MAPS WITH LABEL-DRIVEN DIFFUSION

**Harel Mendelman, Ronen Talmon**
Viterbi Faculty of Electrical and Computer Engineering, Technion
`harel.men@campus.technion.ac.il, ronen@ee.technion.ac.il`

## ABSTRACT

In this paper, we introduce Supervised Diffusion Maps (SDM) and Semi-Supervised Diffusion Maps (SSDM), which transform the well-known unsupervised dimensionality reduction algorithm, Diffusion Maps, into supervised and semi-supervised learning tools. The proposed methods, SDM and SSDM, are based on our new approach that treats the labels as a second view of the data. This unique framework allows us to incorporate ideas from multi-view learning. Specifically, we propose constructing two affinity kernels corresponding to the data and the labels. We then propose a multiplicative interpolation scheme of the two kernels, whose purpose is twofold. First, our scheme extracts the common structure underlying the data and the labels by defining a diffusion process driven by the data and the labels. This label-driven diffusion produces an embedding that emphasizes the properties relevant to the label-related task. Second, the proposed interpolation scheme balances the influence of the two kernels. We show on multiple benchmark datasets that the embedding learned by SDM and SSDM is more effective in downstream regression and classification tasks than existing unsupervised, supervised, and semi-supervised nonlinear dimension reduction methods.

## 1 INTRODUCTION

Manifold learning has emerged as a powerful approach for uncovering the underlying structure of complex high-dimensional datasets. Aiming to mitigate the "curse of dimensionality", the core idea behind manifold learning is the manifold assumption, i.e., assuming that high-dimensional data lies on or near a lower-dimensional manifold, which captures the essential features of the data. This lower-dimensional representation can reveal patterns, relationships, and intrinsic geometries obscured in the original ambient high-dimensional space.

Manifold learning techniques are designed to discover these low-dimensional manifolds by leveraging the geometric properties of the data. Unlike linear dimensionality reduction methods, such as Principal Component Analysis (PCA) (Pearson, 1901), manifold learning algorithms are non-linear and can model more complex structures. These methods are particularly effective when data is generated by non-linear processes, making them well-suited for applications in many applied science fields, such as bioinformatics, biomedical, and neuroscience (Diaz-Papkovich et al., 2021; Sulam et al., 2017; Dimitriadis et al., 2018).

Even with recent advances in deep learning, manifold learning methods remain relevant for their interpretability, effectiveness with small datasets, lower risk of overfitting, and lower computational requirements. Additionally, their concepts have influenced geometric deep learning, with ongoing research promising further advancements in that area.

Numerous manifold learning algorithms have been developed in the last two decades. Some of the most prominent techniques include Isomap (Tenenbaum et al., 2000), Locally Linear Embedding (LLE) (Roweis & Saul, 2000), Laplacian Eigenmaps (LE) (Belkin & Niyogi, 2003), Diffusion Maps (DM) (Coifman & Lafon, 2006), t-Distributed Stochastic Neighbor Embedding (t-SNE) (Van der Maaten & Hinton, 2008), and UMAP (McInnes et al., 2018).

Despite their advantages, standard manifold learning methods have limitations such as parameter selection, noise sensitivity, and computational complexity in large-scale datasets, which have already

been partially resolved in recent advances, e.g., Karoui & Wu (2016); Shen & Wu (2022). In this work, we focus on another limitation – the incorporation of label information. At their core, manifold learning methods are unsupervised, and therefore, they often do not provide an effective solution that exploits all the information in a large body of problems where some label information exists.

In this paper, we present Supervised Diffusion Maps (SDM) and Semi-Supervised Diffusion Maps (SSDM), which are supervised and semi-supervised variants of Diffusion Maps (Coifman & Lafon, 2006). Unlike existing supervised and semi-supervised manifold learning methods (See Section 2), we present a new approach, where we view the labels as an additional data modality and employ concepts of multimodal manifold learning. Concretely, we propose to construct a separate affinity kernel for the labels, allowing us to capture the underlying geometry of the labels in addition to that of the data. Then, based on Alternating Diffusion (AD) (Lederman & Talmon, 2018), a multimodal manifold learning method that relies on a product of affinity kernels, we propose a multiplicative kernel interpolation scheme of the data and label kernels. This results in a kernel representing a transition probability matrix of a Markov chain on the data and labels. In analogy to AD (Talmon & Wu, 2019), this kernel approximates a continuous diffusion process on the manifold consisting of a two-step diffusion process: first on the labels and then on the data, henceforth referred to as "label-driven diffusion". This approach reveals the common structure underlying the data and labels, resulting in a data embedding that emphasizes only the properties relevant to the specified task related to the labels. We present theoretical justification and experimental results that show the advantages of the proposed SDM and SSDM compared to existing manifold learning methods on several benchmarks.

Our main contributions are as follows:

- **Multi-View Approach to Supervised and Semi-Supervised Manifold Learning**: Viewing labels as a second source of information.

- **New Kernel Interpolation Scheme**: Interpolating affinity kernels to reveal their commonalities, while providing a mechanism for weighting the contribution of each.

- **Experimental Results**: Showing that SDM and SSDM outperform existing non-linear manifold learning algorithms on real-world benchmark datasets.

## 2 RELATED WORK

Numerous supervised and semi-supervised adaptations of manifold learning algorithms have been explored. These methods generally adopt one of the following approaches.

Several supervised and semi-supervised variants of manifold learning algorithms, such as LLE (Zhang, 2009), Diffusion Maps (Szlam et al., 2008), and t-SNE (Hajderanj et al., 2019), modify the similarity or distance metric. Typically, the dissimilarity metric is redefined as follows:

$$D(x_i, x_j) = \begin{cases} \sqrt{1 - \exp\left(\frac{-d^2(x_i, x_j)}{\epsilon}\right)}, & \text{if } y_i = y_j \\ \sqrt{\exp\left(\frac{d^2(x_i, x_j)}{\epsilon}\right) - \alpha}, & \text{if } y_i \neq y_j \end{cases}, \tag{1}$$

where $x_i$ and $x_j$ are the data samples, $y_i$ and $y_j$ are their corresponding labels, $d(\cdot, \cdot)$ is a distance metric, and $\epsilon$ and $\alpha$ are tunable parameters.

Another approach optimizes inter-class and intra-class objectives, as employed for example by a variant of Isomap (Yang et al., 2016), where the goal is to preserve intra-class distances while increasing inter-class distances:

$$\min_{\mathbf{Z}} \alpha \sum_{y_i = y_j} \left(d(x_i, x_j) - d(z_i, z_j)\right)^2 - \beta \sum_{y_i \neq y_j} d^2(x_i, x_j), \tag{2}$$

where $\mathbf{Z}$ represents a lower-dimensional embedding, and $\alpha$ and $\beta$ control the trade-off between the intra-class and inter-class objectives.

In addition, some methods like variants of LE (Ma et al., 2019) and UMAP (McInnes et al., 2018), divides the objective into two components, one that focuses on the data and another that incorporates

label information. An example of such an objective function is:

$$\min_{\mathbf{Z}} \alpha \sum \left(d(x_i, x_j) - d(z_i, z_j)\right)^2 + \beta \sum d^2(z_i, \widehat{z}_c), \tag{3}$$

where $\widehat{z}_c$ serves as the representative point for class $c$.

These three approaches often overlook the potential similarity between the geometric structures underlying the data and the labels, as they integrate label information into the data manifold, which may result in a skewed representation. Recognizing that labels and data might reside on similar structures, our method introduces a new approach by learning a representation that captures the commonalities between the geometries of the data and labels.

## 3 BACKGROUND

### 3.1 DIFFUSION MAPS

Diffusion Maps, introduced by Coifman & Lafon (2006), present spectral analysis-based low-dimensional data embedding with diffusion geometry and diffusion distance.

Consider a measure space $(\mathcal{M}, d\mu)$, where $\mathcal{M}$ is a smooth Riemannian manifold and $d\mu(x) = p(x)dx$ is a measure with density $p(x) \in C^3(\mathcal{M})$. Assume $\mathcal{M}$ is isometrically embedded in $\mathbb{R}^d$, and let $\{x_i \in \mathcal{M}\}_{i=1}^n \subset \mathbb{R}^d$ be a set of $n$ points sampled from $p(x)$.

Diffusion Maps begin with constructing an affinity matrix $\mathbf{W}$, typically using a Gaussian kernel:

$$\mathbf{W}(i,j) = \exp\left(-\frac{d(x_i, x_j)^2}{\epsilon}\right), \tag{4}$$

where $d(\cdot, \cdot)$ is a distance metric, and $\epsilon$ is the kernel scale. This matrix captures local pairwise similarities between the sampled points. Then, normalization of $\mathbf{W}$ is applied using two diagonal matrices, $\mathbf{D}_1$ and $\mathbf{D}_2$, derived from the sums of rows of $\mathbf{W}$ and $\widetilde{\mathbf{K}} = \mathbf{D}_1^{-1}\mathbf{W}\mathbf{D}_1^{-1}$, respectively. The resulting normalized matrix $\mathbf{K} = \mathbf{D}_2^{-1}\widetilde{\mathbf{K}}$ is a row-stochastic matrix, viewed as the transition probability matrix of a Markov chain on the data (Coifman & Lafon, 2006).

From the eigenvectors $v_k$ and eigenvalues $\mu_k$ of $\mathbf{K}$, Diffusion Maps is defined by $\Psi_k(x_i) = \mu_k^\tau v_k(i)$, where $\tau > 0$ denotes the diffusion time parameter, and $k = 1, 2, \ldots, n$. It was shown in Coifman & Lafon (2006) that the matrix $\mathbf{K}$ represents a diffusion process on the continuous underlying manifold when a large number of points is available. The diffusion propagation of a sample $i$ is modeled by iteratively applying the matrix $\mathbf{K}$. Specifically, if $\delta_i$ denotes a one-hot vector with a single non-zero entry at the $i$-th coordinate representing an initial mass concentrated at sample $i$, the sequence of diffused mass vectors is given by $\delta_i^\top \mathbf{K}^\tau$. The diffusion distance between samples $i$ and $j$ is defined by $d_\tau(i,j) = \|\delta_i^\top \mathbf{K}^\tau - \delta_j^\top \mathbf{K}^\tau\|_2$. This distance metric not only captures the direct similarities between samples but also accounts for their connectivity throughout the Markov chain. As a result, it provides a robust measure that is less sensitive to local variations, outliers, and noise within the dataset. In addition, it can be approximated by the Euclidean distance between the respective Diffusion Maps, i.e., $\|\Psi_k(x_i) - \Psi_k(x_j)\|_2$.

### 3.2 ALTERNATING DIFFUSION

Alternating Diffusion (AD) (Lederman & Talmon, 2018) is an extension of Diffusion Maps designed to extract the common structure of two aligned datasets $\{x_i^{(1)}\}_{i=1}^n$ and $\{x_i^{(2)}\}_{i=1}^n$. AD starts by building affinity matrices, $\mathbf{W}^{(1)}$ and $\mathbf{W}^{(2)}$, for each dataset as in Eq. 4. These affinity matrices are normalized to form the diffusion operators $\mathbf{K}^{(1)}$ and $\mathbf{K}^{(2)}$, following the procedure used in Diffusion Maps (See Subsection 3.1). The AD operator is defined by the product of the two diffusion operators $\mathbf{K}^{(1)\cap(2)} = \mathbf{K}^{(1)}\mathbf{K}^{(2)}$. Like Diffusion Maps, this operator defines a propagation that consists of alternating diffusion steps via $\mathbf{K}^{(1)}$ and $\mathbf{K}^{(2)}$ on the two datasets. Lederman & Talmon (2018) showed that the resulting diffusion process captures the common structure between the two datasets, minimizing the influence of dataset-specific factors.

## 4 PROPOSED METHOD

Consider a labeled training dataset $\{(x_i, y_i)\}_{i=1}^n$ consisting of $n$ data samples $x_i \in \mathbb{R}^d$ and their corresponding labels $y_i$, and a test dataset $\{\overline{x}_j\}_{j=1}^m$ consisting of $m$ unlabeled test data samples $\overline{x}_j \in \mathbb{R}^d$. Our goal is to obtain an informative low dimensional representation of both the training and test datasets, given by the embedding $\{\Psi(x_i)\}_{i=1}^n$ and $\{\Psi(\overline{x}_j)\}_{j=1}^m$ of the samples into a Euclidean space $\mathbb{R}^\ell$, $\ell < d$. Same as in many dimension reduction methods, e.g., Belkin & Niyogi (2003); Coifman & Lafon (2006), we aim to find an embedding, whose Euclidean distances are meaningful. Specifically, following Diffusion Maps (Coifman & Lafon, 2006), we build an embedding, whose Euclidean distances approximate the diffusion distances (see Subsection 3.1). Importantly, such a construction assumes unlabeled data. The main novelty in this paper is that we propose to account for the given labels of the training data, such that samples with similar labels are mapped to close points in the embedding space, and samples with dissimilar labels are mapped to distant points, thereby improving the embedding, especially for downstream tasks such as classification and regression.

Seemingly, the desired embedding described above could be obtained directly using an appropriate manipulation of the classical Multidimensional Scaling (MDS) (Carroll & Arabie, 1998) or other supervised and semi-supervised dimensionality reduction techniques that minimize distances between similar labels and maximize distances between dissimilar labels (See Section 2). However, this requires unnatural data and label weighting and usually has poor scalability (Ma et al., 2019). We propose a different approach that mitigates these shortcomings. Our approach relies on the premise that both the data and the labels have similar underlying geometric structures, and consequently, uncovers their commonality through a diffusion process that integrates both data and labels.

For simplicity, we first describe SDM (supervised setting) in the context of a single unlabeled test sample $\overline{x}$. We then extend this description to multiple unlabeled test samples $\{\overline{x}_j\}_{j=1}^m$, and finally describe SSDM (semi-supervised settting). The first step is to build an affinity matrix $\mathbf{W}_D$ in $\mathbb{R}^{(n+1) \times (n+1)}$ on the set $\{x_i\}_{i=1}^n \cup \overline{x}$, consisting of the training data and one test sample. The elements of this affinity matrix are given by

$$\mathbf{W}_D(i,j) = \exp\left(-\frac{d_D^2(x_i, x_j)}{\epsilon_D}\right), \tag{5}$$

where $d_D(\cdot, \cdot)$ is a distance metric in $\mathbb{R}^d$, e.g., the Euclidean distance, and $\epsilon_D$ is a hyperparameter. Then, the affinity matrix is normalized twice, as described in Subsection 3.1, resulting in a data kernel, denoted by $\mathbf{D} \in \mathbb{R}^{(n+1) \times (n+1)}$.

Similarly, we propose to build an affinity matrix between the labels. Since the label of the test sample $\overline{x}$ is missing, we define the following affinity matrix, whose elements are given by:

$$\mathbf{W}_P(i,j) = \begin{cases} \exp\left(-\frac{d_P^2(y_i, y_j)}{\epsilon_P}\right), & \text{if } i, j \leq n \\ 0, & \text{if } i > n \text{ or } j > n, \text{ and } i \neq j \\ 1, & \text{if } i, j > n \text{ and } i = j \end{cases}, \tag{6}$$

where $d_P(\cdot, \cdot)$ is a distance metric between the labels and $\epsilon_P$ is a hyperparameter. If the labels are continuous (regression problem), we compute the distance between the labels using a standard distance metric, e.g., the Euclidean distance. If the labels are discrete (classification problem), we propose the following metric (other metrics may also be considered):

$$d_P(y_i, y_j) = \begin{cases} \frac{1}{n_l \cdot n_m} \sum_{y_k \in \mathcal{C}_l, y_q \in \mathcal{C}_m} d_D(x_k, x_q), & \text{if } y_i \in \mathcal{C}_l, y_j \in \mathcal{C}_m, l \neq m \\ 0, & \text{if } y_i = y_j \end{cases}, \tag{7}$$

where $\mathcal{C}_l = \{y \mid y \in \text{Class } l\}$ for $l = 1, \ldots, C$, $C$ is the number of classes, and $n_l$ is the size of $\mathcal{C}_l$. As described in Subsection 3.1, the affinity matrix $\mathbf{W}_P$ is normalized twice, giving rise to the label kernel $\mathbf{P} \in \mathbb{R}^{(n+1) \times (n+1)}$, representing the *prior* label information we have on the training dataset. Specifically, the construction in Eq. 6 adds an isolated node representing $\overline{x}$ to the transition graph $\mathbf{P}$.

Once the kernels $\mathbf{D}$ and $\mathbf{P}$ are obtained, the goal is to find embeddings for $\{(x_i, y_i)\}_{i=1}^n$ and $\overline{x}$. To exploit that both the data and labels share similar underlying geometric structures, we build on AD (Lederman & Talmon, 2018) and utilize the product of kernels to uncover commonalities. Specifically, we propose the following interpolation scheme:

$$\mathbf{\Gamma}(t) = \mathbf{P}^{1-t}\mathbf{D}^t, \quad 0 \leq t \leq 1. \tag{8}$$

In this scheme, $\boldsymbol{\Gamma}(t)$ denotes the interpolated kernel, where $t$ balances between $\mathbf{P}$ and $\mathbf{D}$. The contribution of our method extends beyond the introduction of a label kernel. It enhances AD in two notable ways. First, it accommodates kernels with partial alignment as $\overline{x}$ lacks a corresponding label (Eq. 6). Second, it provides a mechanism for weighting the contribution of the data and the labels through the hyperparameter $t$. In Section 5, we show that $\boldsymbol{\Gamma}(t)$ facilitates an approximation of the embedding that would have been obtained in a fully aligned setting without missing labels.

We use $\boldsymbol{\Gamma}(t)$ as a kernel in a manner similar to Diffusion Maps (See Subsection 3.1) to obtain the embedding of $\{x_i\}_{i=1}^n$ and $\overline{x}$ for $t$. Specifically, the embedding is given by

$$\Psi_k(x_i) = \mu_k v_k(i), \quad \Psi_k(\overline{x}) = \mu_k v_k(n+1), \tag{9}$$

where $k = 1, 2, \ldots, n+1$ indexes the components, $\mu_k$ denotes the $k$-th eigenvalue and $v_k(i)$ denotes the $i$-th entry of the $k$-th right eigenvector of $\boldsymbol{\Gamma}(t)$. We summarize the key steps of SDM with a single unlabeled test sample in Algorithm 1.

Since the embedding of $\overline{x}$ is generated without alignment with a label, we observed a slight improvement in downstream tasks by introducing a similar distortion to the embedding of the training set, ensuring better consistency with the embedding of $\overline{x}$. For details, see Appendix B.

When multiple test samples $\{\overline{x}_j\}_{j=1}^m$ are given, we apply the described procedure sequentially for each sample. This results in the embedding of both the training and test samples, given by:

$$\{\Psi_k(x_i)\}_{i=1}^n, \quad \{\Psi_k(\overline{x}_j)\}_{j=1}^m. \tag{10}$$

A summary of SDM with multiple unlabeled test samples is provided in Algorithm 2 in Appendix A.1.

---

**Algorithm 1** Generate the embedding of $\{x_i\}_{i=1}^n \cup \overline{x}$

---

1: Construct $\mathbf{W}_D$ (Eq. 5)
2: Construct $\mathbf{W}_P$ (Eq. 6)
3: Obtain $\mathbf{D}$, $\mathbf{P}$ by normalizing $\mathbf{W}_D$, $\mathbf{W}_P$ as in Sec. 3.1
4: Compute $\boldsymbol{\Gamma}(t)$ for $t \in [0, 1]$ (Eq. 8)
5: Obtain $\{\Psi_k(x_i)\}_{i=1}^n, \Psi_k(\overline{x})\}$ (Eq. 9)

---

Considering one test sample at a time neglects possible informative mutual relationships between the test samples and is computationally intensive. Instead, we propose to enhance the embedding of multiple test samples by considering a *semi-supervised* setting consisting of the labeled training data and all the unlabeled test data simultaneously. This will also expedite SDM, alleviating the need to construct and interpolate two affinity kernels for each training and test sample.

Concretely, instead of adding a single unlabeled sample to the training data, we append the entire test set, resulting in the union $\{x_i\}_{i=1}^n \cup \{\overline{x}_j\}_{j=1}^m$. Then, we build the data kernel $\mathbf{D}$ and label kernel $\mathbf{P}$ that correspond to this union set based on the affinities Eq. 5 and Eq. 6, respectively, and the dual normalization, where the kernel matrices are now in $\mathbb{R}^{(n+m)\times(n+m)}$. This construction adds $m$ isolated nodes to the transition graph $\mathbf{P}$, unlike the single isolated node added in SDM. We obtain the embedding of all training and test samples using a single kernel $\boldsymbol{\Gamma}(t)$. Specifically, the embedding is given by $\Psi_k(x_i) = \mu_k v_k(i)$ for $i = 1, 2, \ldots, n$ and $\Psi_k(\overline{x}_j) = \mu_k v_k(n + j)$ for $j = 1, 2, \ldots, m$. We term this algorithm SSDM, which results, as in SDM, in embedding as in Eq. 10. Since SSDM uses a single pair of affinity kernels for the entire dataset, its runtime is significantly shorter compared to SDM, as demonstrated in Section 6.2. A summary of SSDM is provided in Algorithm 3 in Appendix A.1.

## 5 Theoretical Justification

In this section, to simplify the analysis, we consider the equally weighted interpolation, denoted by $\mathbf{PD}$, instead of the tunable form $\mathbf{P}^{1-t}\mathbf{D}^t$ (refer to Appendix C.4 for details on the tunable form). Additionally, for the purpose of the analysis, we define the *inaccessible* kernel representing affinities between the training labels and the unknown test label in the set $\{y_i\}_{i=1}^n \cup \overline{y}$ by $\mathbf{L}$. Analogous to $\mathbf{P}$, we first compute the affinity kernel $\mathbf{W}_L$ as follows:

$$\mathbf{W}_L(i, j) = \exp\left(-\frac{d_P^2(y_i, y_j)}{\epsilon_P}\right),$$

for $i, j = 1, \ldots, n+1$, and then, apply the same normalization to obtain the *inaccessible* kernel $\mathbf{L} \in \mathbb{R}^{(n+1)\times(n+1)}$. In practice, we use $\mathbf{P}$ as a proxy for the inaccessible $\mathbf{L}$.

Let $\mathcal{N}_i^{(D)}$ denote the $\delta_D$-neighborhood of $x_i$, given by $\mathcal{N}_i^{(D)} = \{x_j \mid d_D^2(x_i, x_j) < \delta_D\}$. Similarly, let $\mathcal{N}_i^{(L)}$ denote the $\delta_L$-neighborhood of $y_i$, defined as $\mathcal{N}_i^{(L)} = \{y_j \mid d_P^2(y_i, y_j) < \delta_L\}$. We assume that the values of $\mathbf{W}_D$ and $\mathbf{W}_L$ outside the $\delta_D$-neighborhood and $\delta_L$-neighborhood, respectively, are negligible.

**Proposition 1.** *The absolute value of the difference between any element* $(i, j)$ *of the inaccessible kernel product* $\mathbf{LD}$ *and the corresponding element* $(i, j)$ *of the available* $\mathbf{PD}$ *is bounded as follows:*

$$|[\mathbf{LD}]_{i,j} - [\mathbf{PD}]_{i,j}| \leq \begin{cases} \frac{1}{|\mathcal{N}_i^{(L)}||\mathcal{N}_{n+1}^{(D)}|} & 1 \leq i \leq n \,, y_i \in \mathcal{N}_{n+1}^{(L)}, x_j \in \mathcal{N}_{n+1}^{(D)} \\ \max\left\{r_j, \frac{1}{|\mathcal{N}_{n+1}^{(D)}|}\right\} & i = n+1 \,, x_j \in \mathcal{N}_{n+1}^{(D)} \\ 0 & otherwise \end{cases} \,,$$

*where* $r_j = \min\left\{\frac{1}{|\mathcal{N}_{n+1}^{(L)}|}, \frac{1}{|\mathcal{N}_j^{(D)}|}\right\}$ *and* $|\mathcal{N}_i^{(D)}|$ *denotes the size of the set, representing the number of samples within the neighborhood of the $i$-th sample.*

**Corollary 1.** *If the neighborhoods* $\mathcal{N}_i^{(D)}$ *and* $\mathcal{N}_i^{(L)}$ *of each sample* $i$ *have at least* $N_1$ *neighbors, the bounds in Proposition 1 simplify to:*

$$|[\mathbf{LD}]_{i,j} - [\mathbf{PD}]_{i,j}| \leq \begin{cases} \frac{1}{N_1^2} & 1 \leq i \leq n \,, y_i \in \mathcal{N}_{n+1}^{(L)}, x_j \in \mathcal{N}_{n+1}^{(D)} \\ \frac{1}{N_1} & i = n+1 \,, x_j \in \mathcal{N}_{n+1}^{(D)} \\ 0 & otherwise \end{cases} \,,$$

See Appendix C.2 for the proof of Proposition 1.

Based on the elementwise similarity of $\mathbf{PD}$ and $\mathbf{LD}$, we show that the eigenvectors of $\mathbf{PD}$ approximate the eigenvectors of $\mathbf{LD}$, utilizing the concept of a pseudo-spectrum.

**Definition 1** ($\epsilon$-pseudo-spectrum (Trefethen, 2020))**.** *Given a matrix* $\mathbf{M} \in \mathbb{R}^{n \times n}$, *the* $\epsilon$-*pseudo-spectrum for a small* $\epsilon > 0$ *is defined by:*

$$\sigma_\epsilon(\mathbf{M}) = \{\mu \in \mathbb{R} \mid \exists v \in \mathbb{R}^n \text{ with } \|v\|_2 = 1 \text{ s.t. } \|(\mathbf{M} - \mu\mathbf{I})v\|_2 \leq \epsilon\},$$

*where* $\sigma(\mathbf{M})$ *denotes the set of eigenvalues of* $\mathbf{M}$, $\mathbf{I}$ *represents the identity matrix, and* $\|\cdot\|_2$ *denotes the $\ell_2$ norm.*

**Proposition 2.** *Let* $v$ *be an eigenvector of* $\mathbf{LD}$ *with a corresponding eigenvalue* $\mu$. *If the neighborhoods* $\mathcal{N}_i^{(D)}$ *and* $\mathcal{N}_i^{(L)}$ *of each sample* $i$ *have at least* $N_1$ *neighbors and at most* $N_2$ *neighbors, then,* $v$ *is a* $\epsilon$-*pseudo-eigenvector of* $\mathbf{PD}$ *with a corresponding* $\epsilon$-*pseudo-eigenvalue* $\mu$, *i.e.,* $\|(\mathbf{PD} - \mu\mathbf{I})v\| \leq \epsilon$, *where* $\epsilon = \sqrt{\frac{N_2^2}{N_1^3} + \frac{N_2}{N_1^2}}$.

**Corollary 2.** *If* $N_1 = N_2 = N$, *then the value of* $\epsilon$ *in Proposition 2 simplifies to* $\epsilon = \sqrt{\frac{2}{N}}$.

See Appendix C.3 for the proof of Proposition 2.

Combining Propositions 1 and 2 indicates that when using $\mathbf{P}$ as a proxy for $\mathbf{L}$, the inaccessible embedding that would have been obtained by $\mathbf{LD}$ is approximated by the proposed embedding based on $\mathbf{PD}$.

We note that the analysis presented in this subsection can be straightforwardly extended to accommodate multiple test samples, as in the case of SSDM. In this extended setting, $\epsilon$ is given by $\epsilon = \sqrt{\frac{c+1}{N}}$, where $c$ represents the maximum number of test samples $j = n+1, \ldots, n+m$ within the neighborhoods $\mathcal{N}_i^{(D)}$ and $\mathcal{N}_i^{(L)}$ of each training sample $i = 1, \ldots, n$.

Next, we exploit the fact that $\mathbf{D}$, $\mathbf{L}$, and $\mathbf{P}$ could be viewed as transition probability matrices of a Markov chain on the dataset (See Subsection 3.1). This viewpoint was extensively studied and exploited in the context of Diffusion Maps (Coifman & Lafon, 2006), where the Markov chain gives rise to a diffusion process on the data. Therefore, $\mathbf{LD}$ and $\mathbf{PD}$ are also transition probability matrices of Markov chains, where each step involves two transitions: the first is on the labels and the

second is on the data, defining "label-driven diffusion". In this analysis, we focus on the $(i, n+1)$-th elements of the matrices, representing the transition probabilities from any labeled sample $i \neq n+1$ to the $(n+1)$-th unlabeled sample. While the $(n+1)$-th node is isolated in $\mathbf{P}$ (i.e., $[\mathbf{P}]_{i,n+1} = 0$), the $(i, n+1)$-th elements of $\mathbf{LD}$ and $\mathbf{PD}$ are given by

$$[\mathbf{LD}]_{i,n+1} = \sum_{j=1}^{n+1} \mathbf{L}_{i,j}\mathbf{D}_{j,n+1}, \quad [\mathbf{PD}]_{i,n+1} = \sum_{j=1}^{n+1} \mathbf{P}_{i,j}\mathbf{D}_{j,n+1} = \sum_{j=1}^{n} \mathbf{L}_{i,j}\mathbf{D}_{j,n+1}, \qquad (11)$$

because $\mathbf{P}_{i,j} = \mathbf{L}_{i,j}$ for $i, j \neq n+1$, and $\mathbf{P}_{i,n+1} = 0$ for $i \neq n+1$.

Thus, the $(n+1)$-th node becomes reachable in $\mathbf{PD}$, while the entries $[\mathbf{LD}]_{i,n+1}$ and $[\mathbf{PD}]_{i,n+1}$ differ only by the last term in the sum, $\mathbf{L}_{i,n+1}\mathbf{D}_{n+1,n+1}$. The term $\mathbf{L}_{i,n+1}\mathbf{D}_{n+1,n+1} \neq 0$ only if the $(n+1)$-th sample is in $\mathcal{N}_i^{(L)}$. According to Corollary 1, the absolute value of this term is bounded by $1/N_1^2$. Therefore, if the $(n+1)$-th sample is not in $\mathcal{N}_i^{(L)}$, the transition probability from the $i$-th node to the $(n+1)$-th node in two diffusion steps through $\mathbf{PD}$ is equal to that through $\mathbf{LD}$. Otherwise, the transition probabilities are approximately equal when $N_1$ is large. Appendix C.5 provides an example comparing label-driven diffusion to ordinary two-step diffusion solely on the data.

## 6 EXPERIMENTAL RESULTS

We present experimental results that showcase the performance of SDM and SSDM on a synthetic dataset and 12 real datasets. These experiments demonstrate the effectiveness of SDM and SSDM in learning low-dimensional embeddings and their impact on downstream tasks, such as regression and classification, compared to several classical and recent baselines. We provide a Python implementation of SDM and SSDM.[1]

### 6.1 TOY PROBLEM

We generated a dataset similar to the toy problem presented in Lederman & Talmon (2018). Our dataset consists of 500 images with three figures: Superman, Spider-Man, and Flash. In each image, the three figures are rotated by different degrees. The images are of size $100 \times 100$ pixels (i.e., 10,000-dimensional data). Each image is labeled with Superman's rotation angle, where the angles of the other two figures are viewed as nuisance factors. The dataset is divided into a 50% training set and a 50% testing set, where the labels of the testing set are disregarded. Figure 1 shows an example of an image with a label of 54° (the angle of Superman). Our toy task is to reveal the rotation angle of Superman in unlabeled images of the three figures.

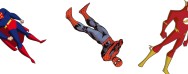

This dataset is designed to embody an explicit underlying manifold structure. The rotation angle of each figure could be viewed as a point on a one-dimensional sphere $\mathcal{S}^1$, such that each image, containing three independent angles, lies on the product manifold $\mathcal{S}^1 \times \mathcal{S}^1 \times \mathcal{S}^1$. The eigenvalues and eigenfunctions of the Laplace-Beltrami operator of $\mathcal{S}^1$ are (Gallier, 2009):

Figure 1: Superman at a 54-degree rotation.

$$\mu_k = -k^2, \quad v_k^{(1)} = \cos(kx_i), \quad v_k^{(2)} = \sin(kx_i), \qquad (12)$$

providing closed-form analytical expressions for the spectral components that are computed in our method in a data-driven manner.

We applied SDM and SSDM to this toy problem. We used the Euclidean distance in the data kernel $\mathbf{D}$ and angular distance in the label kernel $\mathbf{P}$ to accommodate the cyclic nature of the angles.

Figure 2 shows the evolution of the principal spectral component denoted by $\Psi_1^{(t)}$ from the label kernel $\mathbf{P} = \mathbf{\Gamma}(t=0)$ to the data kernel $\mathbf{D} = \mathbf{\Gamma}(t=1)$ obtained by SDM. In Appendix D.1, we present additional components (Figure 11) and SSDM components (Figure 12). The x-axis represents the angle (label), and the y-axis represents the value of the respective entry of $\Psi_1^{(t)}$, where blue and red points indicate training and test samples, respectively. Note that each point in Figure 2 (SDM) was generated using a different kernel, while the points in Figure 12 in Appendix D.1 (SSDM) were generated using a single kernel, as described in Section 4. At $t = 0$, we see that the entries of $\Psi_1^{(0)}$

---

[1]The code is available at https://github.com/harel147/sdm.

P (Label Kernel)
t=0.0    t=0.02    t=0.3    t=0.8    D (Data Kernel)
t=1.0

Figure 2: Progression of $\Psi_1^{(t)}$ from $t = 0$ (label kernel) to $t = 1$ (data kernel). The x-axis represents the angle (i.e., the label) and the y-axis the value of the corresponding $\Psi_1^{(t)}$ entry, where blue and red points for training (labeled) and test (unlabeled) samples, respectively.

are meaningless, since $\mathbf{\Gamma}(t = 0)$ contains no information about the unlabeled samples. At $t = 1$, we see that the entries of $\Psi_1^{(1)}$ are highly noisy, as $\mathbf{\Gamma}(t = 1)$ does not incorporate any label information, and the results coincide with that of the unsupervised Diffusion Maps. For $0 < t < 1$, we see that the entries of $\Psi_1^{(t)}$ are cleaner and more similar to the spherical harmonic functions in Eq. 12, especially as $t$ decreases and $\mathbf{\Gamma}(t)$ approaches the label kernel. To complement the presentation, in Figure 13 in Appendix D.1 we present the two-dimensional embedding of the toy dataset obtained using SSDM with various $t$ values. We display results for both the toy problem dataset and a baseline dataset containing only images of Superman (without the nuisance figures). This visualization demonstrates that SSDM generates informative embeddings of $\mathcal{S}^1$ that are consistent with the embedding obtained by Diffusion Maps applied to the "clean" data (without the nuisance figures).

We evaluated SDM and SSDM by training a KNN regressor on the top four spectral components for each $t$ in $0, 0.01, \ldots, 1.0$ using the training set and assessing performance on the test set using angular Mean Absolute Error (MAE).

Figure 3 displays the MAE of SDM and SSDM in predicting Superman's rotation angle as a function of $t$, compared to the baseline performance of unsupervised Diffusion Maps (DM). The results demonstrate the superiority of SDM and SSDM over DM due to the incorporation of labels. Specifically, SDM achieves the best results for small values of $t$, while SSDM performs better for larger values of $t$ in a wide range, highlighting its robustness to the tuning of the hyperparameter $t$ in addition to its superior computational efficiency.

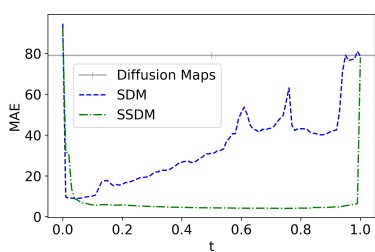

Figure 3: Evaluation of SDM and SSDM on the toy problem.

To demonstrate the importance of the proposed interpolation scheme, we compared it to a simpler interpolation method analogous to the one employed by AD (Lederman & Talmon, 2018). Table 2 in Appendix D.2 illustrates the effectiveness of SDM and SSDM compared to this simpler approach.

To observe the effect of increasing the ratio of labeled samples in the semi-supervised setting, we analyzed SSDM performance of with varying labeled data ratios. Consistent training (100 samples, 20% of the dataset) and testing sets (100 samples) were used across experiments to ensure that the results reflect only embedding quality. The minimal MAE for the optimal $t$ is 17.42° for a 20% labeled data ratio, 5.84° for 40%, 4.76° for 60%, and 4.46° for 80%. These results demonstrate the improved effectiveness of SSDM with increased labeled data. Notably, performance only slightly worsens when decreasing from 80% labels to 40%, and significantly deteriorates only when the ratio is reduced to 20%. Figure 14 in Appendix D.1 shows the MAE as a function of $t$ for each ratio.

## 6.2 REAL DATASETS

### 6.2.1 SUPERVISED SETTING

We evaluated 7 datasets from the UCI Machine Learning Repository and Scikit-learn library (see Appendix E.1 for details). Each dataset underwent 50 data splits (70% training - 30% testing) for

Table 1: Evaluation results, reporting NMSE for regression ('R') and Misclassification Rate for classification ('C'). Our algorithms are S/SDM for supervised SDM and semi-supervised SSDM.

| Dataset | | Unsupervised Algorithms | | | | | | Semi/Supervised Algorithms | | |
|---|---|---|---|---|---|---|---|---|---|---|
| Name | Type | UMAP | Isomap | tSNE | LE | LLE | DM | S/SUMAP | SStSNE | S/SDM |
| **Supervised Setting** | | | | | | | | | | |
| Iris | C | 0.037 | 0.047 | 0.044 | **0.034** | 0.06 | 0.051 | 0.035 | 0.041 | **0.034** |
| Ionosphere | C | 0.15 | 0.112 | 0.117 | 0.12 | 0.121 | 0.108 | 0.155 | 0.111 | **0.073** |
| Arrhythmia | C | 0.51 | 0.484 | 0.476 | 0.443 | 0.533 | 0.506 | 0.485 | 0.456 | **0.431** |
| Musk | C | 0.191 | 0.189 | 0.145 | 0.122 | 0.204 | 0.18 | 0.171 | 0.163 | **0.101** |
| Yacht | R | 0.549 | 0.395 | 0.451 | 0.208 | 0.498 | 0.43 | 0.549 | - | **0.12** |
| Boston | R | 0.084 | 0.087 | 0.074 | 0.083 | 0.08 | 0.096 | 0.083 | - | **0.068** |
| Liver | R | 0.508 | 0.50 | 0.511 | 0.481 | 0.502 | 0.497 | 0.519 | - | **0.474** |
| **Semi-Supervised Setting** | | | | | | | | | | |
| Mice | C | 0.029 | 0.03 | 0.01 | 0.045 | **0.002** | 0.162 | 0.029 | 0.047 | 0.016 |
| Rice | C | 0.146 | 0.127 | 0.148 | 0.152 | 0.156 | 0.161 | 0.153 | - | **0.104** |
| Silhouettes | C | 0.39 | 0.35 | 0.378 | 0.419 | 0.297 | 0.437 | 0.389 | 0.315 | **0.248** |
| Raisin | C | 0.211 | 0.195 | 0.216 | 0.216 | 0.204 | 0.26 | 0.22 | 0.18 | **0.176** |
| Concrete | R | 0.077 | 0.066 | 0.064 | 0.08 | 0.051 | 0.085 | 0.07 | - | **0.036** |

learning dimension reduction embedding and training KNN models—5 neighbors for regression and 1 neighbor for classification.

For each dataset and algorithm, we reduced the original data dimensionality to a range of 1 to 30 dimensions. We then trained KNN models and calculated the errors – Normalized Mean Square Error (NMSE) for regression and Misclassification Rate for classification. These errors were averaged across the 50 data splits for each dataset. We report the minimum average error achieved (within the range of 1 to 30 dimensions) and the corresponding standard deviation.

We compared the proposed SDM against several unsupervised algorithms: Diffusion Maps (Jiang & Shen, 2020), UMAP (McInnes et al., 2018), Isomap (Tenenbaum et al., 2000), t-SNE (Van der Maaten & Hinton, 2008), Laplacian Eigenmaps (LE) (Belkin & Niyogi, 2003), and Locally Linear Embedding (LLE) (Roweis & Saul, 2000). We also compared SDM against supervised UMAP (SUMAP) (McInnes et al., 2018) and semi-supervised t-SNE (SStSNE) (McInnes et al., 2016). We compared SDM against semi-supervised t-SNE rather than supervised t-SNE because, to the best of our knowledge, a public implementation of a supervised variant of t-SNE is not available.

To ensure robustness, we selected $t$ for SDM once on the first data split and applied it consistently across all 50 splits. A similar approach was used for selecting the hyperparameters of the other methods, namely, the number of neighbors in UMAP, SUMAP, Isomap, LE, and LLE, and the perplexity in t-SNE and SStSNE. For SStSNE, we do not show regression results as the method does not support continuous labels. See Appendix B for additional implementation details.

Table 1 (top part) presents the results. For the complete table, including standard deviations and with the number of dimensions that yielded the smallest error indicated in parentheses, see Table 4 in Appendix E.2. The best-performing result for each dataset among all algorithms is in bold. We see that our SDM consistently outperforms unsupervised Diffusion Maps across all datasets. Additionally, our SDM achieves the best results for all datasets.

### 6.2.2 Semi-Supervised Setting

We considered five datasets from the UCI Machine Learning Repository and the Scikit-learn library. Since SSDM computes a single kernel for the entire dataset, it is faster and facilitates application to larger datasets (with more than 500 samples) than those used in the supervised setting. Each dataset underwent the same processing and evaluation procedure as in the supervised setting, including generating 50 data splits and training KNN models.

We compared our SSDM against the same unsupervised algorithms as in the supervised setting. To adhere to the semi-supervised setting, we learned the embedding on the entire dataset rather than on

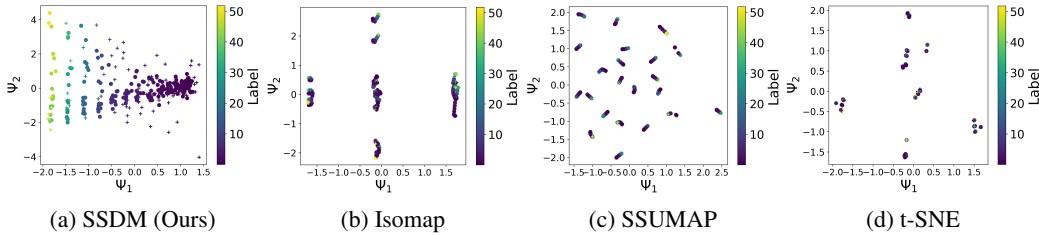

(a) SSDM (Ours)  (b) Isomap  (c) SSUMAP  (d) t-SNE

Figure 4: Two-dimensional embedding of the Yacht dataset: dots ($\cdot$) for labeled samples and pluses (+) for unlabeled samples.

the train set. Additionally, we compared our SSDM against the semi-supervised variants of UMAP (SSUMAP) (McInnes et al., 2018) and t-SNE (SStSNE) (McInnes et al., 2016). As in the supervised setting, we used the same approach for selecting the hyperparameters: $t$ for our SSDM and the number of neighbors and perplexity for the other algorithms. SStSNE excludes the Rice dataset due to convergence issues and the Concrete dataset because it does not support regression.

Table 1 (bottom part) presents the results. For the complete table, see Table 4 in Appendix E.2. We see in the table that our SSDM consistently outperforms the unsupervised Diffusion Maps across all datasets. Moreover, our SSDM achieves the best result for all datasets except for Mice.

In Figure 4, we present the two-dimensional embedding of the Yacht dataset obtained by SSDM, Isomap, SSUMAP, and t-SNE. In the embedding generated by SSDM, we see that the labels (represented by the color) correspond to the 2D location in the embedded space, in contrast to the other embeddings, thereby demonstrating the effectiveness of our method in visually representing the data and capturing the label information.

### 6.3 Complexity and Runtime Analysis

The time complexity of SDM is $O(n^4)$, with a space complexity of $O(n^2)$, where $n$ is the dataset size. This makes SDM impractical for large datasets. In contrast, SSDM has a time complexity of $O(n^3)$ while maintaining a space complexity of $O(n^2)$, making it more suitable for larger datasets. Table 6 in Appendix E.5 compares the runtimes of the evaluated algorithms. While SDM is slower than other methods, SSDM achieves runtimes comparable to the alternatives.

In Appendix F, we propose an optimized SSDM variant with a time complexity of $O(k^2 \cdot n)$ and a space complexity of $O(k \cdot n)$, where $k$ is the number of randomly sampled labeled points. We evaluate its performance on six large datasets, as shown in Table 7 (Appendix F). Runtime comparisons in Table 8 (Appendix F) show that the optimized SSDM is significantly faster than SSUMAP across all datasets. A detailed complexity analysis is provided in Appendix F.

### 7 Conclusion

In this paper, we introduce Supervised Diffusion Maps (SDM) and Semi-Supervised Diffusion Maps (SSDM), which extend the classical Diffusion Maps algorithm by incorporating label information. Treating labels as an additional view and using a multiplicative interpolation of affinity kernels, our methods effectively fuse and balance between the structures underlying data and labels. Results on benchmark datasets demonstrate that SDM and SSDM give rise to low-dimensional representations that lead to superior performance in downstream regression and classification tasks compared to existing methods, showcasing their effectiveness in leveraging label information for enhanced data representation. We remark that the main limitation of SDM is its computational load, as two kernels must be constructed and interpolated for each sample. This limitation was mitigated in SSDM at the expense of higher approximation error between the available partially aligned kernel and the inaccessible fully aligned kernel due to the smaller ratio between labeled and unlabeled samples, leading to a less informative embedding. In future work, we plan to explore the utility and adaptation of the fusion of data and label kernels, developed in this paper, in the context of the emerging geometric deep learning.

ETHICS STATEMENT

This paper introduces new machine learning techniques. All datasets used in this study are publicly available and commonly employed in benchmarking machine learning algorithms. The methods developed respect the privacy and integrity of the data, and no personal or sensitive information was used.

REPRODUCIBILITY STATEMENT

The details of the experimental settings are provided in Section 6. We include detailed proofs of the theoretical analysis in Appendix C. Additional implementation details can be found in Appendix B. Our source code is available at `https://github.com/harel147/sdm`.

ACKNOWLEDGMENTS

The work of HM and RT was supported by the European Union's Horizon 2020 research and innovation programme under grant agreement No. 802735-ERC-DIFFOP.

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

# A  ALGORITHMS AND BLOCK DIAGRAMS

## A.1  ALGORITHMS

---

**Algorithm 2** Supervised Diffusion Maps (SDM)

---

**Input:**
- Train data: $n$ original dimension train data samples $\{x_i\}_{i=1}^{n}$
- Test data: $m$ original dimension test data samples $\{\overline{x}_j\}_{j=1}^{m}$
- Train labels: $\{y_i\}_{i=1}^{n}$

**Parameters:**
- Interpolation parameter: $t \in (0, 1)$
- Data distance metric: metric used to measure the distance between data samples
- Labels distance metric: metric used to measure the distance between labels

**Output:** Low-dimensional embeddings of the training and testing data

**Step 1: Extract embeddings along the interpolation for the test set**

1: Construct $\mathbf{W}_P$ (Eq. 6). Obtain $\mathbf{P}$ by normalizing $\mathbf{W}_P$ as in 3.1.
2: For each $\overline{x}_j$ in the test set:
    a. Construct $\mathbf{W}_D$ for $\{x_i\}_{i=1}^{n} \cup \overline{x}_j$ (Eq. 5). Obtain $\mathbf{D}$ by normalizing $\mathbf{W}_D$ as in 3.1.
    b. Calculate $\mathbf{\Gamma}(t)$ (Eq. 8).
    c. Obtain $\Psi_k(\overline{x}_j)\}$ (Eq. 9).

**Step 2: Extract embeddings along the interpolation for the train set**

1: For each $(x_i, y_i)$ in the training set, treat it as unlabeled by ignoring $y_i$, and apply the procedure described in Step 1 to collect the embeddings of $x_i$.

---

---

**Algorithm 3** Semi-Supervised Diffusion Maps (SSDM)

---

**Input:**
- Train data: $n$ original dimension train data samples $\{x_i\}_{i=1}^{n}$
- Test data: $m$ original dimension test data samples $\{\overline{x}_j\}_{j=1}^{m}$
- Train labels: $\{y_i\}_{i=1}^{n}$

**Parameters:**
- Interpolation parameter: $t \in (0, 1)$
- Data distance metric: metric used to measure the distance between data samples
- Labels distance metric: metric used to measure the distance between labels

**Output:** Low-dimensional embeddings of the training and testing data

1: Construct $\mathbf{W}_D$ for $\{x_i\}_{i=1}^{n} \cup \{\overline{x}_j\}_{j=1}^{m}$ (Eq. 5). Obtain $\mathbf{D}$ by normalizing $\mathbf{W}_D$ as in 3.1.
2: Construct $\mathbf{W}_P$ (Eq. 6). Obtain $\mathbf{P}$ by normalizing $\mathbf{W}_P$ as in 3.1.
3: Calculate $\mathbf{\Gamma}(t)$ (Eq. 8).
4: Obtain $\{\Psi_k(x_i)\}_{i=1}^{n}$ and $\{\Psi_k(\overline{x}_j)\}_{j=1}^{m}$, where $\Psi_k(x_i) = \mu_k v_k(i)$ for $i = 1, 2, \ldots, n$ and $\Psi_k(\overline{x}_j) = \mu_k v_k(n+j)$ for $j = 1, 2, \ldots, m$. Here, $\mu_k$ denotes the $k$-th eigenvalue and $v_k(i)$ denotes the $i$-th entry of the $k$-th right eigenvector of $\mathbf{\Gamma}(t)$.

---

## A.2 BLOCK DIAGRAMS

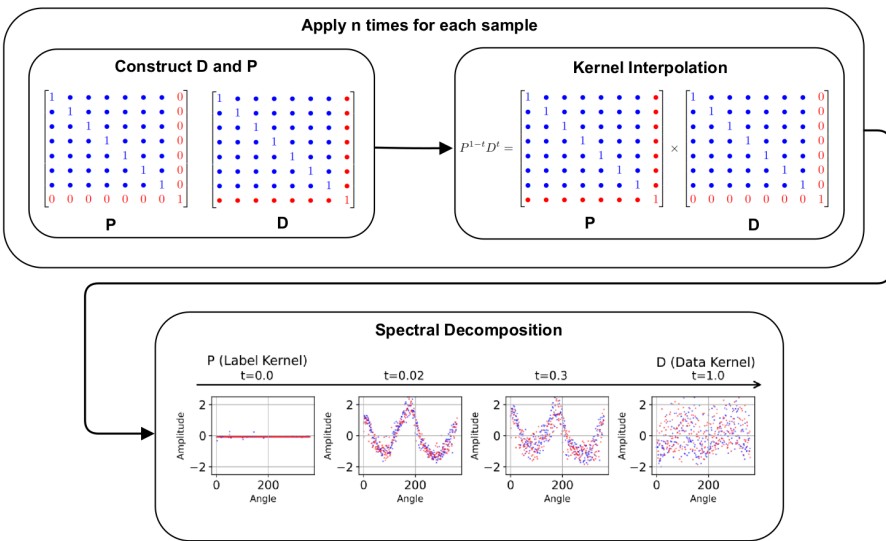

Figure 5: Block diagram of the SDM framework. The data kernel $\mathbf{D}$ and label kernel $\mathbf{P}$ are constructed separately for each sample. The embeddings shown in the lower part represent the eigenvectors. Blue rows/columns in $\mathbf{D}$ and $\mathbf{P}$ correspond to labeled data samples, while the red row/column correspond to the unlabeled data sample. Similarly, the blue embeddings correspond to labeled data, and the red embeddings correspond to unlabeled data samples.

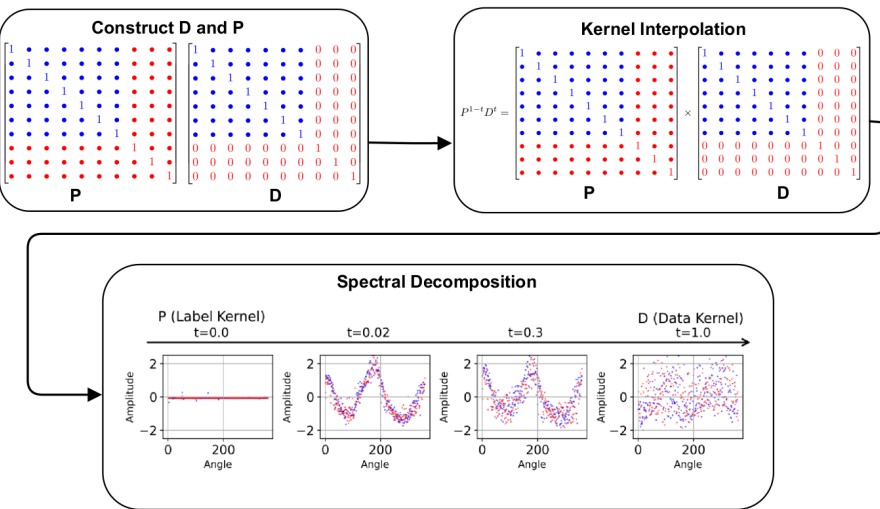

Figure 6: Block diagram of the SSDM framework. The data kernel $\mathbf{D}$ and label kernel $\mathbf{P}$ are constructed once for all samples. The embeddings shown in the lower part represent the eigenvectors. Blue rows/columns in $\mathbf{D}$ and $\mathbf{P}$ correspond to labeled data samples, while red rows/columns correspond to unlabeled data samples. Similarly, the blue embeddings correspond to labeled data, and the red embeddings correspond to unlabeled data samples.

# B  ADDITIONAL IMPLEMENTATION DETAILS

In our study, we utilized the official Sklearn implementations for t-SNE, Isomap, LE and LLE. For UMAP, supervised UMAP, and semi-supervised UMAP we employed the official implementations. Diffusion Maps were implemented by us.

Regarding supervised t-SNE, although multiple versions have been published in academic papers, the code implementations were not released. Therefore, we utilized the semi-supervised t-SNE version (McInnes et al., 2016) implemented by Leland McInnes, the creator of UMAP (McInnes et al., 2018).

For all algorithms, we primarily used default parameters, except for selecting the number of neighbors parameter for UMAP, supervised UMAP, semi-supervised UMAP, Isomap, LE, and LLE, and the perplexity for t-SNE and semi-supervised t-SNE.

To select the $\epsilon$ for the Gaussian kernels of the data and label kernel for SDM and SSDM, we applied the following heuristic: we calculated 16 kernels using different $\epsilon$ values from $10^{-5}$ to $10^{10}$, and chose the $\epsilon$ value that resulted in the kernel with the largest number of eigenvalues in the range $[0.0001, 0.9999]$. We applied this heuristic because we observed numerical issues when the eigenvalues of the kernel were too large or too small. We did not optimize over $\epsilon$ to try to improve performance on the datasets.

We selected the flow resolution parameter, which is the number of sampled kernels along the interpolation, defining $\{t_i\}_{i=1}^{l}$ in the interval $[0, 1]$, to be 101 for all evaluated datasets. This means that we generated embeddings for each test and train sample along the interpolation for all kernels at $t_i$ in $0, 0.01, 0.02, \ldots, 0.99, 1.0$, then selected the optimal $t$ from this set based on the first data split, as described in Subsection 6.2.1. As for the distance metric, we used only Euclidean distance for both the data kernel and the label kernel.

As for SSDM, the optimal $t$ is usually large ($[0.9, 1)$). This means that the interpolated kernel result is close to $\mathbf{D}$. We found it effective to apply the following SVD shrinkage-based denoising tactic to the kernel $\mathbf{D}$ before applying SSDM. We reconstruct $\mathbf{D}$ with singular values sampled from a sigmoid normalized between 0 and 1. Practically, we do this as follows:

---

**Algorithm 4** Pseudocode for SVD Shrinkage-Based Denoising Tactic

---

1: $U, \Sigma, V \leftarrow SVD(\mathbf{D})$
2: $s \leftarrow \text{Linspace}(-5, 5, n)$
3: $\sigma \leftarrow \frac{1}{1+\exp(s)}$
4: $\sigma \leftarrow \frac{\sigma-\min(\sigma)}{\max(\sigma)-\min(\sigma)}$
5: $\Sigma \leftarrow \sigma$
6: $\mathbf{D} \leftarrow U\Sigma V$

---

As described in 4 for SDM, the embedding of the training set for $t$ can be obtained straightforwardly using $\Psi_k(x_i) = \mu_k v_k(i)$ for $i = 1, 2, \ldots, n$. However, since the embedding of $\bar{x}$ was generated without alignment with a label, we observed a slight improvement in downstream tasks by applying a similar distortion to the training set embeddings, ensuring greater consistency with the embedding of $\bar{x}$. To achieve this, we extend the $\bar{x}$ procedure to all training samples $\{x_i\}_{i=1}^{n}$ using a leave-one-out approach, where each sample $(x_i, y_i)$ in the training set is treated as unlabeled. In this case, we construct kernels in $\mathbb{R}^{n \times n}$ using the data samples $\{x_j\}_{j \neq i}^{n} \cup x_i$ and the $(n-1)$ labels $\{y_j\}_{j \neq i}^{n}$. For each sample $x_i$, we compute the embedding as $\Psi_k(x_i) = \mu_k v_k(n)$.

In the original Diffusion Maps algorithm, Eigenvalue Decomposition (EVD) is applied to the kernel. In our approach, we utilize Singular Value Decomposition (SVD) instead, as previously suggested for Alternating Diffusion (AD) in Talmon & Wu (2019). We chose SVD because it produces results similar to EVD, but with the added advantages of faster computation and greater numerical stability in NumPy's implementation.

The hardware utilized for testing the runtimes of SDM and SSDM, as depicted in Table 6 in Appendix E.5, is the ROG Strix G16 Asus laptop equipped with an Intel i9-14900HX processor.

## C  THEORETICAL JUSTIFICATION: PROOFS AND EXPLANATIONS

### C.1  DISCUSSION OF SHARED STRUCTURE ASSUMPTION

Our method is designed to extract the shared underlying structure between the data and labels, relying on the assumption that they exhibit a common geometry. While this assumption is challenging to validate directly, we propose an alternative, more practical criterion: when the labels of two samples are close (i.e., the label distance is small), the distances produced by our label-driven diffusion process should be smaller than the direct pairwise distances between the data samples. This leads to a higher transition probability in our kernel compared to the data kernel. Conversely, if the labels are distant (i.e., the label distance is large), the distances from our label-driven diffusion process should exceed the direct pairwise distances between data samples, resulting in a lower transition probability in our kernel compared to the data kernel. If this relationship holds, it suggests that the assumption is valid for the given dataset. In simpler terms, the assumption is that our label-driven diffusion more accurately reflects the label distances than the direct pairwise distances between the data samples. This assumption can be formalized as follows:

**Assumption 1.** *For any two data samples $i$ and $j$, if the labels $y_i$ and $y_j$ are similar, i.e., $d_P(y_i, y_j) < \eta$, then $[PD]_{i,j} > [D]_{i,j}$. Conversely, if the labels $y_i$ and $y_j$ are not similar, i.e., $d_P(y_i, y_j) > \eta$, then $[PD]_{i,j} < [D]_{i,j}$,*

where $\eta$ is a parameter that defines whether two labels are considered similar or not.

When this assumption does not hold – that is, when the direct pairwise distances between data samples align better with the label distances than our diffusion distances – the traditional Diffusion Maps is expected to outperform our method. In such cases, the kernel $D$ will better represent the label distances than our kernel $PD$.

To assess whether the assumption holds for real datasets, we propose using heatmap visualizations of both our method's kernel and the unsupervised Diffusion Maps kernel. These heatmaps, sorted by label with rows representing unlabeled samples and columns representing labeled samples, provide a visual means of evaluating the assumption. An ideal heatmap for classification should exhibit diagonal high-value blocks corresponding to each class, indicating that samples have high transition probabilities to others within the same class and low probabilities to those in other classes, as outlined in Assumption 1.

Figures 7 and 8 show heatmaps for our method and the unsupervised Diffusion Maps applied to the Mice dataset (Higuera & Cios, 2015) and the Raisin dataset (Çinar & Tasdemir, 2023), respectively. In both cases, our kernel better approximates the ideal diagonal block structure compared to the unsupervised Diffusion Maps kernel. This result aligns with the Misclassification Rates reported in Table 4. For the Mice dataset, our SSDM embedding achieves a Misclassification Rate of $0.016 \pm 0.009$, while the unsupervised Diffusion Maps yield a Misclassification Rate of $0.162 \pm 0.019$. Similarly, for the Raisin dataset, SSDM achieves a Misclassification Rate of $0.176 \pm 0.019$, compared to $0.26 \pm 0.021$ for the unsupervised Diffusion Maps.

Conversely, Figure 9 presents heatmaps for the Customers dataset (Cardoso, 2013), where neither method effectively captures the label structure. The heatmaps lack the expected diagonal block patterns, reflecting a failure to align with the assumption. This is further reflected in the Misclassification Rates: SSDM achieves $0.328 \pm 0.044$, while the unsupervised Diffusion Maps perform slightly better with a Misclassification Rate of $0.283 \pm 0.047$. In this case, where the assumption does not hold, as evident in Figure 9, the unsupervised Diffusion Maps outperform SSDM.

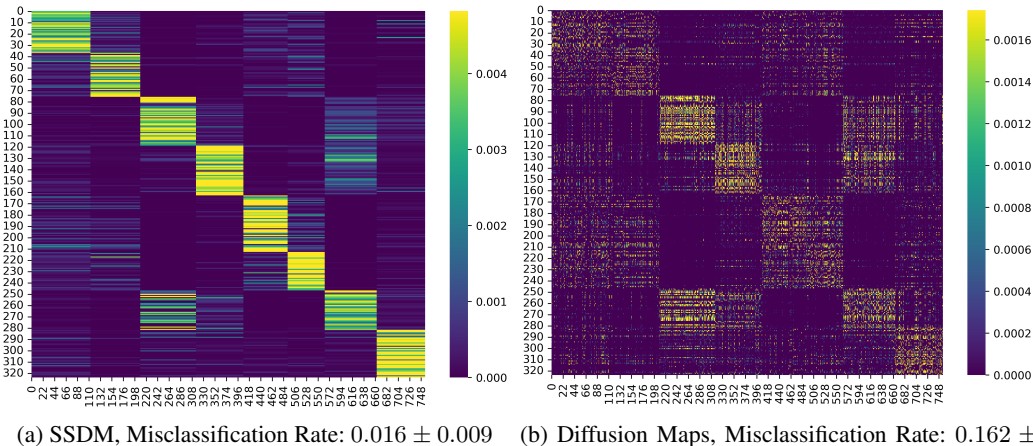

(a) SSDM, Misclassification Rate: $0.016 \pm 0.009$

(b) Diffusion Maps, Misclassification Rate: $0.162 \pm 0.019$

Figure 7: Heatmap visualization of our method's kernel and the unsupervised Diffusion Maps kernel for the Mice dataset.

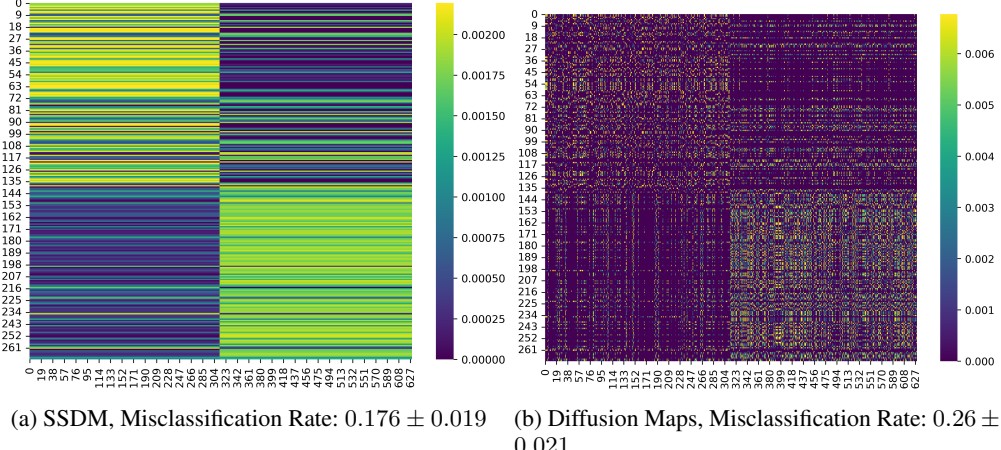

(a) SSDM, Misclassification Rate: $0.176 \pm 0.019$

(b) Diffusion Maps, Misclassification Rate: $0.26 \pm 0.021$

Figure 8: Heatmap visualization of our method's kernel and the unsupervised Diffusion Maps kernel for the Raisin dataset.

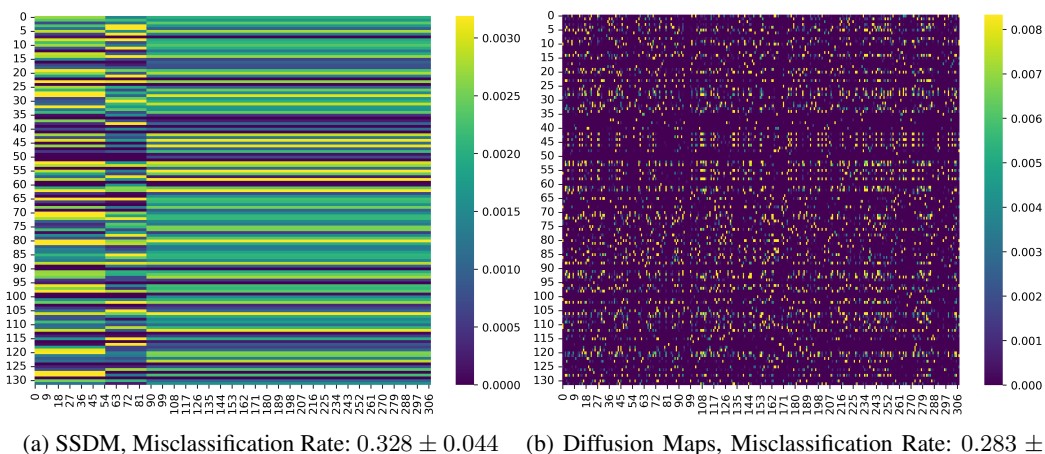

(a) SSDM, Misclassification Rate: $0.328 \pm 0.044$

(b) Diffusion Maps, Misclassification Rate: $0.283 \pm 0.047$

Figure 9: Heatmap visualization of our method's kernel and the unsupervised Diffusion Maps kernel for the Customers dataset.

## C.2 PROOF OF PROPOSITION 1

***Proof*** *of Proposition 1.* We consider modifications for tractability. First, we clip negligible affinity values by

$$\mathbf{W}_D(i,j) = \begin{cases} \exp\left(-\frac{d_D^2(x_i,x_j)}{\epsilon_D}\right), & \text{if } d_D^2(x_i,x_j) \leq \delta_D \\ 0, & \text{if } d_D^2(x_i,x_j) > \delta_D \end{cases}. \tag{13}$$

Second, we consider a different kernel normalization given by $\mathbf{D} = \mathbf{D}_1^{-1}\mathbf{W}_D$, where $\mathbf{D}_1$ is a diagonal matrix consisting of the sum of rows of $\mathbf{W}_D$. Note that the resulting normalized data kernel $\mathbf{D}$ is row-stochastic. In addition, it was shown in Coifman & Lafon (2006) that it is similar to the symmetric kernel used so far when the data is uniformly distributed on the data manifold. The label kernel $\mathbf{P}$ and the inaccessible kernel $\mathbf{L}$ undergo the same modifications.

Let $\mathbf{L}_{i,j}$ denote the elements of $\mathbf{L}$ and $\mathbf{D}_{i,j}$ denote the elements of $\mathbf{D}$. We refer to the $i$-th row vector of $\mathbf{L}$ as $\mathbf{l_i}$ and the $i$-th column vector of $\mathbf{D}$ as $\mathbf{d_i}$. Using this notation, we can express the elements of $\mathbf{PD}$ and $\mathbf{LD}$ as follows:

$$[\mathbf{LD}]_{i,j} = \mathbf{l_i d_j}, \tag{14}$$

$$[\mathbf{PD}]_{i,j} = \begin{cases} \mathbf{l_i d_j} - \mathbf{L}_{i,n+1}\mathbf{D}_{n+1,j}, & \text{for } 1 \leq i \leq n \\ \mathbf{D}_{n+1,j}, & \text{if } i = n+1 \end{cases}. \tag{15}$$

Thus, we can relate $\mathbf{LD}$ and $\mathbf{PD}$ as $\mathbf{E} = \mathbf{LD} - \mathbf{PD}$, where the elements of $\mathbf{E}$ are defined by:

$$\mathbf{E}_{i,j} = \begin{cases} \mathbf{L}_{i,n+1}\mathbf{D}_{n+1,j}, & \text{for } 1 \leq i \leq n \\ \mathbf{l_{n+1}d_j} - \mathbf{D}_{n+1,j}, & \text{if } i = n+1 \end{cases}. \tag{16}$$

Upper bounding $\|\mathbf{E}_{i,j}\|$ is equivalent to upper bounding $\|[\mathbf{LD}]_{i,j} - [\mathbf{PD}]_{i,j}\|$, which is the goal of the proof. Therefore, from now on, we will focus on that.

Next, we define $w_D(x_i)$ and $w_L(y_i)$ as the row sums of $\mathbf{W}_D(i,j)$ and $\mathbf{W}_L(i,j)$. We assume that for sufficiently large $n$, the row sums of $\mathbf{L}$ and $\mathbf{P}$ are approximately the same, as $\mathbf{P}$ before row normalization only lacks the values of the last column of $\mathbf{L}$ before row normalization. We bound $w_D(x_i)$ as follows:

$$w_D(x_i) = \sum_{x_j \in \mathcal{N}_i^{(D)}} \exp\left(-\frac{d_D{}^2(x_i,x_j)}{\epsilon_D}\right) \geq \sum_{x_j \in \mathcal{N}_i^{(D)}} \exp\left(-\frac{\delta_D}{\epsilon_D}\right) \overset{\dagger}{=} |\mathcal{N}_i^{(D)}| \exp(-1) \approx |\mathcal{N}_i^{(D)}|, \tag{17}$$

where $|\mathcal{N}_i^{(D)}|$ denotes the size of the set $\mathcal{N}_i^{(D)}$, and the transition marked by $\dagger$ is achived under the assumption that $\delta_D = \Theta(\epsilon_D)$

Similarly, we get that:

$$w_L(y_i) \geq |\mathcal{N}_i^{(L)}|, \tag{18}$$

where $|\mathcal{N}_i^{(L)}|$ denotes the size of the set $\mathcal{N}_i^{(L)}$.

Thus, based on Eq. 17 and Eq. 18, we can upper bound each element of $D$ and $L$ as follows:

$$\mathbf{D}_{i,j} = \frac{\mathbf{W}_D(i,j)}{w_D(x_i)} \leq \frac{\mathbf{W}_D(i,j)}{|\mathcal{N}_i^{(D)}|} \leq \frac{1}{|\mathcal{N}_i^{(D)}|}, \tag{19}$$

$$\mathbf{L}_{i,j} = \frac{\mathbf{W}_L(i,j)}{w_L(y_i)} \leq \frac{\mathbf{W}_L(i,j)}{|\mathcal{N}_i^{(L)}|} \leq \frac{1}{|\mathcal{N}_i^{(L)}|}. \tag{20}$$

From Eq. 20 and Eq. 19, we obtain:

$$\mathbf{L}_{i,j}\mathbf{D}_{k,l} \leq \frac{1}{|\mathcal{N}_i^{(L)}||\mathcal{N}_k^{(D)}|} \quad \forall i,j,k,l \in [1,n+1]. \tag{21}$$

Now, if we examine the $n$ upper rows of $\mathbf{E}$, they take the form $\mathbf{L}_{i,n+1}\mathbf{D}_{n+1,j}$. Therefore, by Eq. 21, we have:

$$\mathbf{L}_{i,n+1}\mathbf{D}_{n+1,j} \leq \frac{1}{|\mathcal{N}_i^{(L)}||\mathcal{N}_{n+1}^{(D)}|} \overset{\star}{=} \frac{1}{N_1^2} \quad \forall i \in [1,n], \forall j \in [1,n+1]. \tag{22}$$

The transition marked by $\star$ in Eq. 22 is achieved under the assumption that every sample has at least $N_1$ samples in its $\delta$-neighborhood in both $\mathbf{D}$ and $\mathbf{L}$. Similarly, all subsequent transitions in the proof marked with $\star$ are based on the same assumption.

Considering the $(n+1)$-th row of $\mathbf{E}$, it takes the form $\mathbf{l_{n+1}d_j} - \mathbf{D}_{n+1,j}$. Therefore, we can bound it as follows:

$$\|\mathbf{l_{n+1}d_j} - \mathbf{D}_{n+1,j}\| \le \max\{\mathbf{l_{n+1}d_j}, \mathbf{D}_{n+1,j}\}. \tag{23}$$

Considering only $\mathbf{l_{n+1}d_j}$, we can upper bound it by plugging in Eq. 21 as follows:

$$\mathbf{l_{n+1}d_j} = \sum_{i=1}^{n+1} \mathbf{L}_{n+1,i}\mathbf{D}_{i,j} = \sum_{y_i \in \mathcal{N}_{n+1}^{(L)}, x_i \in \mathcal{N}_j^{(D)}} \mathbf{L}_{n+1,i}\mathbf{D}_{i,j} \le \frac{\min\{|\mathcal{N}_{n+1}^{(L)}|, |\mathcal{N}_j^{(D)}|\}}{|\mathcal{N}_{n+1}^{(L)}||\mathcal{N}_j^{(D)}|} \tag{24}$$

$$= \min\left\{\frac{1}{|\mathcal{N}_{n+1}^{(L)}|}, \frac{1}{|\mathcal{N}_j^{(D)}|}\right\} \stackrel{\star}{=} \frac{1}{N_1}. \tag{25}$$

Considering only $\mathbf{D}_{n+1,j}$, we can upper bound it by plugging in Eq. 19 as follows:

$$\mathbf{D}_{n+1,j} \le \frac{1}{|\mathcal{N}_{n+1}^{(D)}|} \stackrel{\star}{=} \frac{1}{N_1}. \tag{26}$$

Therefore, we can conclude that:

$$\|\mathbf{l_{n+1}d_j} - \mathbf{D}_{n+1,j}\| \le \max\left\{\min\left\{\frac{1}{|\mathcal{N}_{n+1}^{(L)}|}, \frac{1}{|\mathcal{N}_j^{(D)}|}\right\}, \frac{1}{|\mathcal{N}_{n+1}^{(D)}|}\right\} \stackrel{\star}{=} \frac{1}{N_1}. \tag{27}$$

We proceed by demonstrating when $E_{i,j} = 0$:

- **For $1 \le i \le n$:** The entry $\mathbf{E}_{i,j}$ is nonzero only if $y_i \in \mathcal{N}_{n+1}^{(L)}$ and $x_j \in \mathcal{N}_{n+1}^{(D)}$.

- **For $i = n+1$:** We have
$$\|\mathbf{E}_{i,j}\| = \|\mathbf{l_{n+1}d_j} - \mathbf{D}_{n+1,j}\|,$$
where $\mathbf{l_{n+1}d_j} = \sum_{i=1}^{n+1} \mathbf{L}_{n+1,i}\mathbf{D}_{i,j}$. Therefore,
$$\|\mathbf{E}_{i,j}\| = \|\mathbf{l_{n+1}d_j} - \mathbf{D}_{n+1,j}\| \le \max\{\mathbf{l_{n+1}d_j}, \mathbf{D}_{n+1,j}\}.$$

  Here, $\mathbf{l_{n+1}d_j}$ is nonzero only if there are samples $(x_i, y_i)$ for $i \in [1, n+1]$ such that $y_i \in \mathcal{N}_{n+1}^{(L)}$ and $x_i \in \mathcal{N}_j^{(D)}$. If such samples exist, it implies that the $j$-th sample, for $j \in [1, n+1]$, is close to the $(n+1)$-th sample. For simplicity, we assume this occurs when $y_j \in \mathcal{N}_{n+1}^{(L)}$ and $x_j \in \mathcal{N}_{n+1}^{(D)}$.

  Additionally, $\mathbf{D}_{n+1,j}$ is nonzero only when $x_j \in \mathcal{N}_{n+1}^{(D)}$ for $j \in \{1, \dots, n+1\}$.

  Thus, by combining both cases, $E_{i,j}$ for $i = n+1$ is nonzero only when $x_j \in \mathcal{N}_{n+1}^{(D)}$.

Based on the analysis for the cases when $E_{i,j} = 0$, along with the element-wise bounds provided in Eq. 22 and Eq. 27, we conclude the proof.

$\square$

## C.3 Proof of Proposition 2

***Proof*** *of Proposition 2.* As demonstrated in the **Proof** of Proposition 1, **PD** can be expressed as a perturbed version of **LD** such that $\mathbf{PD} = \mathbf{LD} - \mathbf{E}$, where **E** is defined as follows:

$$\mathbf{E}_{i,j} = \begin{cases} \mathbf{L}_{i,n+1}\mathbf{D}_{n+1,j}, & \text{for } 1 \leq i \leq n \\ \mathbf{l_{n+1}d_j} - \mathbf{D}_{n+1,j}, & \text{if } i = n+1 \end{cases}. \tag{28}$$

By the eigen decomposition, we have $\mathbf{LD}v = \mu v$. Substituting $\mathbf{LD} = \mathbf{PD} + \mathbf{E}$, and noting that $v$ is an eigenvector of **LD** with the corresponding eigenvalue $\mu$, we get:

$$(\mathbf{PD} + \mathbf{E})v = \mu v. \tag{29}$$

We can reorganize this as:

$$(\mathbf{PD} - \mu\mathbf{I})v = -\mathbf{E}v. \tag{30}$$

Applying the norm on both sides, we obtain:

$$\|(\mathbf{PD} - \mu\mathbf{I})v\| = \| - \mathbf{E}v\| = \|\mathbf{E}v\| \leq \|\mathbf{E}\|\|v\| = \|\mathbf{E}\|, \tag{31}$$

where the last equality holds because $\|v\| = 1$.

We proceed by constructing the matrix $\mathbf{E}^{(b)}$ as follows:

$$\mathbf{E}_{i,j}^{(b)} = \begin{cases} \frac{1}{N_1^2}, & \text{for } 1 \leq i \leq n, y_i \in \mathcal{N}_{n+1}^{(L)}, x_j \in \mathcal{N}_{n+1}^{(D)} \\ \frac{1}{N_1}, & \text{if } i = n+1, x_j \in \mathcal{N}_{n+1}^{(D)} \\ 0, & \text{else} \end{cases}. \tag{32}$$

Thus, based on the proof of Proposition 1, $\mathbf{E}^{(b)}$ provides an element-wise bound on **E** by construction, such that $\|\mathbf{E}_{i,j}\| \leq \mathbf{E}_{i,j}^{(b)}$ for all $i, j \in \{1, \ldots, n+1\}$.

We continue by considering $\|\mathbf{E}^{(b)}\|_1$ and $\|\mathbf{E}^{(b)}\|_\infty$, which are computed through the maximal column sum and row sum respectively. We assume that each sample has at most $N_2$ neighbors within its $\delta$-neighborhood in both **D** and **L**. Thus, for each column vector of $\mathbf{E}^{(b)}$, the sum of the first $n$ entries is bounded by $\frac{N_2}{N_1^2}$, and the $(n+1)$-th entry is bounded by $\frac{1}{N_1}$. Consequently, the maximal column sum is bounded by:

$$\|\mathbf{E}^{(b)}\|_1 \leq \frac{N_2}{N_1^2} + \frac{1}{N_1} \overset{\star}{=} \frac{2}{N}. \tag{33}$$

Additionally, the sum of each of the first $n$ row vectors of $\mathbf{E}^{(b)}$ is bounded by $\frac{N_2}{N_1^2}$, while the sum of the $(n+1)$-th row vector is bounded by $\frac{N_2}{N_1}$. Therefore, the maximal row sum is bounded by:

$$\|\mathbf{E}^{(b)}\|_\infty \leq \frac{N_2}{N_1} \overset{\star}{=} 1. \tag{34}$$

The transitions marked by $\star$ in Eq. 33 and Eq. 34 assume that $N_1$ and $N_2$ are approximately equal to $N$, i.e., $N_1 \approx N \approx N_2$.

Using the upper bound of the spectral norm $\|\mathbf{E}^{(b)}\| \leq \sqrt{\|\mathbf{E}^{(b)}\|_1\|\mathbf{E}^{(b)}\|_\infty}$, we get:

$$\|\mathbf{E}^{(b)}\| \leq \sqrt{\|\mathbf{E}^{(b)}\|_1\|\mathbf{E}^{(b)}\|_\infty} \leq \sqrt{\frac{N_2}{N_1}\left(\frac{N_2}{N_1^2} + \frac{1}{N_1}\right)} = \sqrt{\frac{N_2^2}{N_1^3} + \frac{N_2}{N_1^2}} \overset{\star}{=} \sqrt{\frac{2}{N}}. \tag{35}$$

Since the bound of $\|\mathbf{E}^{(b)}\|$ depends on $\|\mathbf{E}^{(b)}\|_1$ and $\|\mathbf{E}^{(b)}\|_\infty$, which are computed through the maximal column sum and row sum respectively, this is also a valid bound for $\|\mathbf{E}\|$, as $\mathbf{E}^{(b)}$ is an element-wise bounding matrix of **E**. Thus, we get:

$$\|\mathbf{E}\| \leq \|\mathbf{E}^{(b)}\| \leq \sqrt{\frac{N_2^2}{N_1^3} + \frac{N_2}{N_1^2}} \overset{\star}{=} \sqrt{\frac{2}{N}}. \tag{36}$$

Assuming that $\epsilon = \sqrt{\frac{N_2^2}{N_1^3} + \frac{N_2}{N_1^2}} \stackrel{\star}{=} \sqrt{\frac{2}{N}}$, and considering that $N$ is sufficiently large, substituting this expression into Eq. 31 yields:

$$\|(\mathbf{PD} - \mu\mathbf{I})v\| = \|\mathbf{E}\| \leq \|\mathbf{E}^{(b)}\| \leq \epsilon. \tag{37}$$

Thus, from **Definition** 1, we conclude that the eigenvector $v$ of $\mathbf{LD}$ is an $\epsilon$-pseudo-eigenvector of $\mathbf{PD}$ with the corresponding eigenvalue $\mu$.

$\square$

### C.4 BALANCING $P$ AND $D$ USING $t$

Considering our interpolation scheme, we have:

$$\mathbf{\Gamma}(t) = \mathbf{P}^{1-t}\mathbf{D}^t, \quad 0 \leq t \leq 1.$$

For any matrix $\mathbf{A} \in \mathbb{R}^{n \times n}$, which are similar to Symmetric Positive Definite (SPD) matrices (such as $\mathbf{P}$ and $\mathbf{D}$ when using row normalization), the fractional power $\alpha$ can be expressed using Eigenvalue Decomposition (EVD) as follows:

$$\mathbf{A}^\alpha = U\Sigma^\alpha V^\top,$$

where $U$ is an orthogonal matrix containing the left eigenvectors of $\mathbf{A}$, $V$ is an orthogonal matrix containing the right eigenvectors of $\mathbf{A}$, and $\Sigma$ is a diagonal matrix with the eigenvalues $\sigma_i$ of $\mathbf{A}$ on its diagonal. The fractional power $\Sigma^\alpha$ is defined as:

$$\Sigma^\alpha = \text{diag}(\sigma_1^\alpha, \sigma_2^\alpha, \ldots, \sigma_n^\alpha),$$

where each eigenvalue $\sigma_i$ is raised to the power $\alpha$.

By raising $\mathbf{P}$ and $\mathbf{D}$ to the fractional powers $1 - t$ and $t$, respectively, we amplify the smaller eigenvalues more significantly. Since smaller eigenvalues correspond to high-frequency components of graph signals, this fractional power operation effectively acts as a high-frequency amplifier. By adjusting $t$, we can control the balance of high-frequency amplification between $\mathbf{P}$ and $\mathbf{D}$. When $t$ is small, the high frequencies of $\mathbf{D}$ are amplified, and when $t$ is large, the high frequencies of $\mathbf{P}$ dominate.

Next, we address the concepts of homophily and heterophily in the context of graphs. Homophily refers to the phenomenon where connected nodes share the same label, whereas in a heterophilic graph, the labels of neighboring nodes can differ. Recent studies (Bo et al., 2021; Chien et al., 2020; Luan et al., 2020) have shown that high-frequency graph signals are empirically effective in tackling the challenges posed by heterophilic graphs. Given that the graph represented by the data kernel $\mathbf{D}$ can be heterophilic as the data may be quite noisy, amplifying the high frequencies of $\mathbf{D}$ using the parameter $t$ can be effective. Similarly, if the quality of the labels is subpar, amplifying the high frequencies of $\mathbf{P}$ may also help. In conclusion, our interpolation scheme balances the amplification of high frequencies in both $\mathbf{P}$ and $\mathbf{D}$, which can be interpreted as interpolation in the frequency domain.

### C.5 LABEL-DRIVEN DIFFUSION ILLUSTRATIVE EXAMPLE

This appendix presents a simple toy problem aimed solely at illustration. Consider a dataset consisting of three samples $\{x_1, x_2, x_3\}$ with corresponding labels $\{y_1, y_2, y_3\}$, and suppose that only $y_1$ and $y_2$ are available and $y_3$ is unavailable. From this dataset, we can construct three graphs with the same node set $\{1, 2, 3\}$, representing the three samples, but with three different transition kernels: $\mathbf{D}$, $\mathbf{L}$, and $\mathbf{P}$.

For the purpose of this analysis, we assume that the transition kernel $\mathbf{L}$ captures the hidden intrinsic geometry of the data as it has access to all the labels (available and unavailable). The transition kernel $\mathbf{P}$ also captures this intrinsic geometry but is incomplete due to the absence of the label $y_3$. The transition kernel $\mathbf{D}$ is noisy, as it is derived from the samples $\{x_1, x_2, x_3\}$ without access to the labels. For simplicity, we construct the graphs without normalization and without self-loops. Consequently, we obtain the graphs and their corresponding transition kernels depicted in Fig. 10.

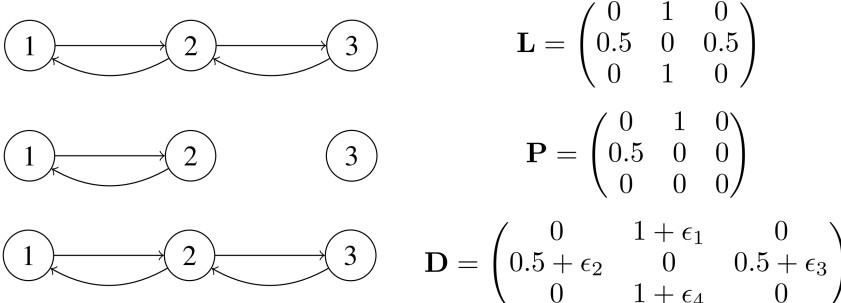

$$\mathbf{L} = \begin{pmatrix} 0 & 1 & 0 \\ 0.5 & 0 & 0.5 \\ 0 & 1 & 0 \end{pmatrix}$$

$$\mathbf{P} = \begin{pmatrix} 0 & 1 & 0 \\ 0.5 & 0 & 0 \\ 0 & 0 & 0 \end{pmatrix}$$

$$\mathbf{D} = \begin{pmatrix} 0 & 1+\epsilon_1 & 0 \\ 0.5+\epsilon_2 & 0 & 0.5+\epsilon_3 \\ 0 & 1+\epsilon_4 & 0 \end{pmatrix}$$

Figure 10: Graphs and transition kernels for $\mathbf{L}$, $\mathbf{P}$, and $\mathbf{P}$.

We now consider a two-step diffusion from node 1 to the unlabeled node 3. The optimal value for this diffusion is given by the $(1, 3)$-th element of $\mathbf{L}^2$, which captures the complete hidden intrinsic structure of the data:

$$\mathbf{L}^2(1, 3) = 0.5$$

If we perform a two-step diffusion on the dataset using the unlabeled samples through $\mathbf{D}^2$ (as in the unsupervised Diffusion Maps), we obtain:

$$\mathbf{D}^2(1, 3) = 0.5 + \epsilon_3 + 0.5\epsilon_1 + \epsilon_1\epsilon_3$$

When employing our method, which involves first step on the labels and second on the data using $\mathbf{PD}$ (where $\mathbf{P}$ serves as a proxy for the inaccessible $\mathbf{L}$), we have:

$$\mathbf{PD}(1, 3) = 0.5 + \epsilon_3$$

Additionally, if we consider two-step diffusion first on the labels and then on the data using the inaccessible $\mathbf{LD}$, which we attempt to approximate with $\mathbf{PD}$, we have:

$$\mathbf{LD}(1, 3) = 0.5 + \epsilon_3$$

Consequently, we observe that the transition probability from node 1 to node 3 using $\mathbf{PD}$ in our method is equal to that using the inaccessible $\mathbf{LD}$. Moreover, the transition probability obtained with $\mathbf{PD}$ exhibits less distortion compared to the optimal transition probability from $\mathbf{L}^2$, while the distortion exhibited by $\mathbf{D}^2$ is larger.

# D    ADDITIONAL EXPERIMENTAL RESULTS FOR THE TOY PROBLEM

## D.1    TOY PROBLEM ADDITIONAL FIGURE

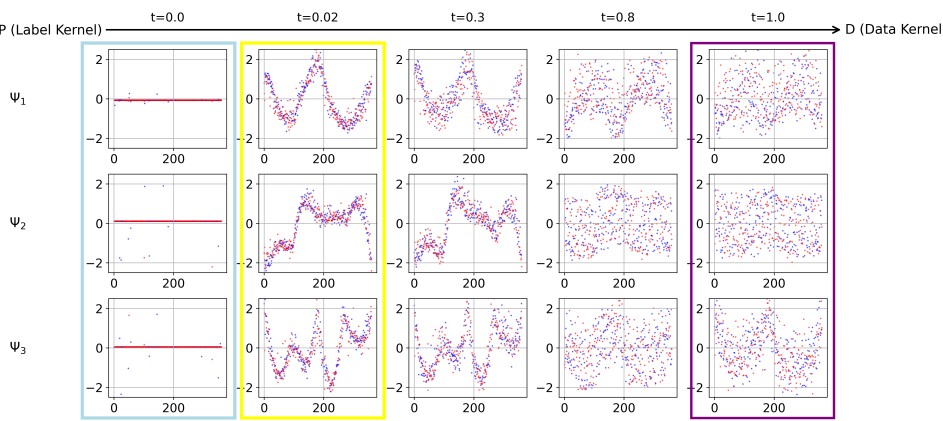

Figure 11: SDM progression of the first three components along the interpolation from the label kernel at $t = 0$ to the data kernel at $t = 1$. Key areas are highlighted: purple rectangle indicate noisy original Diffusion Maps components; yellow rectangle show potential optimal $t$ value components at $t = 0.02$; light blue rectangle enclose non-informative $t = 0$ components.

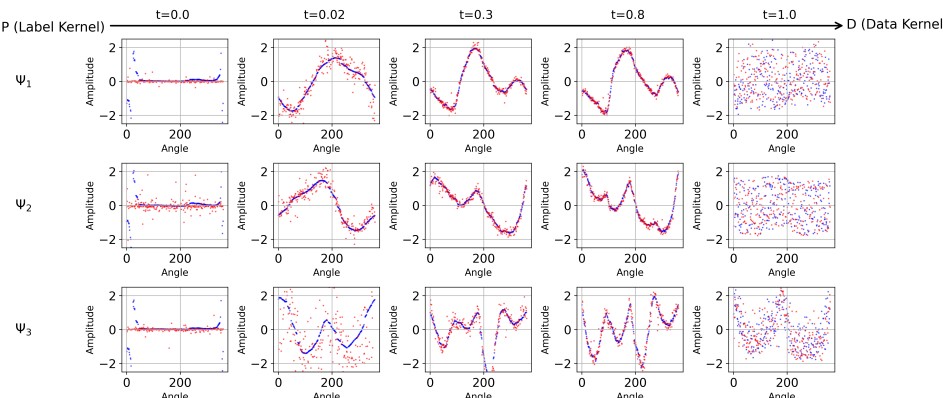

Figure 12: SSDM progression of the first three components along the interpolation from the label kernel at $t = 0$ to the data kernel at $t = 1$. In this case, the optimal $t$ value is large (i.e., 0.8).

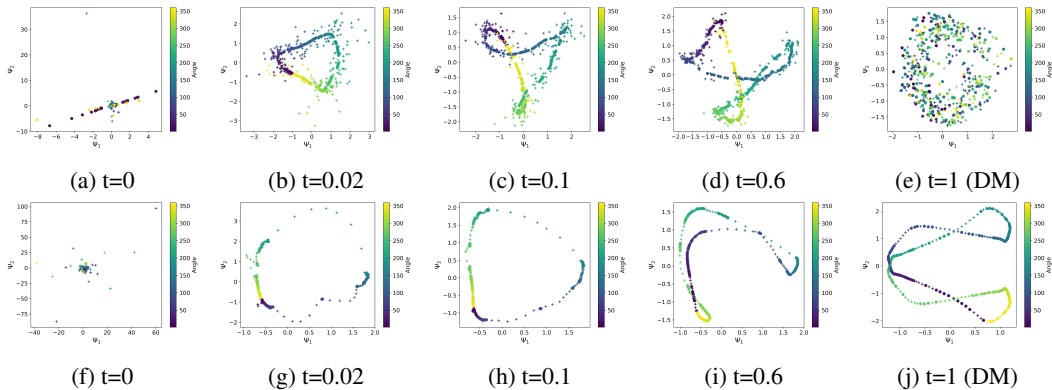

Figure 13: Two-dimensional embedding of SSDM with various $t$ values. The first row displays results for the toy problem dataset, as detailed in Subsection 6.1, featuring three figures (Superman and two interference figures). The second row, serving as a baseline, shows results for a dataset containing only images of Superman, without interference figures. Dots (·) denote labeled training samples, while pluses (+) denote unlabeled test samples.

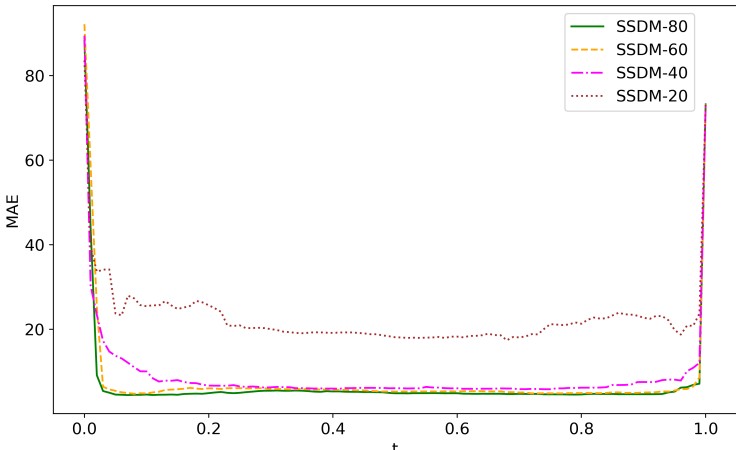

Figure 14: SSDM performance with varying labeled data ratios.

## D.2 COMPARISON WITH AD INTERPOLATION

To demonstrate the importance of the proposed interpolation scheme, we compared it to a simpler interpolation, given by $\mathbf{PD}$, which does not balance between $\mathbf{P}$ and $\mathbf{D}$ using $t$, for both supervised and semi-supervised settings. This simpler interpolation is analogous to the one employed by AD (Lederman & Talmon, 2018).

Table 2 presents the angular MAE and $R^2$ obtained by our SDM and SSDM, as well as by a random guess baseline, DM, and the AD interpolation. The table demonstrates the effectiveness of SDM and SSDM in generating informative embeddings that consistently outperform the other methods. Notably, DM performs only slightly better than a random guess, while AD in the semi-supervised setting is equivalent to a random guess.

Table 2: Evaluation of SDM and SSDM, compared to baselines and AD.

| Method | Supervised | | Semi-supervised | |
|---|---|---|---|---|
| | MAE | $R^2$ | MAE | $R^2$ |
| Random guess | - | - | 90° | 0.0 |
| DM (baseline) | - | - | 79° | 0.08 |
| AD | 50.3° | 0.55 | 85.35° | -0.01 |
| Our Method | 8.88° | 0.97 | 4.1° | 0.99 |

# E  ADDITIONAL DETAILS AND RESULTS FOR REAL DATASETS

## E.1  REAL DATASETS

Table 3 presents the datasets and their properties, where $n$ stands for the number of samples, $d$ for the original data dimensions, and in the 'Type' column, 'C' stands for classification and 'R' for regression.

Table 3: Datasets

| Dataset | Type | $n$ | $d$ |
|---|---|---|---|
| Iris (Fisher, 1988) | C | 150 | 4 |
| Ionosphere (Sigillito & Baker, 1989) | C | 351 | 34 |
| Yacht (Gerritsma & Versluis, 2013) | R | 308 | 6 |
| Boston (Harrison Jr & Rubinfeld, 1978) | R | 506 | 13 |
| Liver (mis, 1990) | R | 345 | 5 |
| Arrhythmia (Guvenir & Quinlan, 1998) | C | 452 | 279 |
| Musk (Chapman & Jain, 1994) | C | 476 | 166 |
| Mice (Higuera & Cios, 2015) | C | 1080 | 77 |
| Rice (mis, 2019) | C | 3810 | 7 |
| Concrete (Yeh, 2007) | R | 1030 | 8 |
| Silhouettes (Mowforth & Shepherd) | C | 845 | 18 |
| Raisin (Çinar & Tasdemir, 2023) | C | 900 | 7 |

## E.2 COMPLETE RESULTS WITH STANDARD DEVIATION

Table 4: Complete evaluation results with standard deviation and with the number of dimensions that yielded the smallest error indicated in parentheses.

| Dataset | | Unsupervised Algorithms | | | | | | Semi/Supervised Algorithms | | |
| Name | Type | UMAP | Isomap | tSNE | LE | LLE | DM | S/SUMAP | SStSNE | S/SDM |
| --- | --- | --- | --- | --- | --- | --- | --- | --- | --- | --- |
| | | | | | **Supervised Setting** | | | | | |
| Iris | C | 0.037(24) ± 0.026 | 0.047(22) ± 0.034 | 0.044(2) ± 0.023 | **0.034(23)** ± 0.027 | 0.06(3) ± 0.048 | 0.051(9) ± 0.047 | 0.035(9) ± 0.022 | 0.041(2) ± 0.035 | **0.034(5)** ± 0.026 |
| Ionosphere | C | 0.15(11) ± 0.029 | 0.112(5) ± 0.028 | 0.117(2) ± 0.026 | 0.12(27) ± 0.025 | 0.121(7) ± 0.031 | 0.108(21) ± 0.025 | 0.155(24) ± 0.027 | 0.111(29) ± 0.028 | **0.073(30)** ± 0.023 |
| Arrhythmia | C | 0.51(23) ± 0.036 | 0.484(25) ± 0.035 | 0.476(4) ± 0.041 | 0.443(28) ± 0.039 | 0.533(22) ± 0.035 | 0.506(29) ± 0.033 | 0.485(15) ± 0.035 | 0.456(30) ± 0.038 | **0.431(10)** ± 0.048 |
| Musk | C | 0.191(26) ± 0.029 | 0.189(23) ± 0.031 | 0.145(2) ± 0.025 | 0.122(18) ± 0.025 | 0.204(30) ± 0.031 | 0.18(16) ± 0.032 | 0.171(25) ± 0.03 | 0.163(7) ± 0.024 | **0.101(8)** ± 0.024 |
| Yacht | R | 0.549(16) ± 0.054 | 0.395(29) ± 0.056 | 0.451(4) ± 0.047 | 0.208(30) ± 0.054 | 0.498(4) ± 0.058 | 0.43(17) ± 0.09 | 0.549(5) ± 0.057 | - | **0.12(2)** ± 0.063 |
| Boston | R | 0.084(7) ± 0.011 | 0.087(16) ± 0.011 | 0.074(10) ± 0.009 | 0.083(30) ± 0.012 | 0.08(13) ± 0.013 | 0.096(6) ± 0.011 | 0.083(15) ± 0.012 | - | **0.068(3)** ± 0.01 |
| Liver | R | 0.508(28) ± 0.051 | 0.50(3) ± 0.051 | 0.511(2) ± 0.051 | 0.481(30) ± 0.046 | 0.502(2) ± 0.058 | 0.497(6) ± 0.051 | 0.519(18) ± 0.06 | - | **0.474(15)** ± 0.054 |
| | | | | | **Semi-Supervised Setting** | | | | | |
| Mice | C | 0.029(30) ± 0.012 | 0.03(28) ± 0.011 | 0.01(10) ± 0.008 | 0.045(13) ± 0.016 | **0.002(30)** ± 0.004 | 0.162(30) ± 0.019 | 0.029(23) ± 0.013 | 0.047(3) ± 0.014 | 0.016(23) ± 0.009 |
| Rice | C | 0.146(9) ± 0.008 | 0.127(27) ± 0.008 | 0.148(7) ± 0.008 | 0.152(8) ± 0.008 | 0.156(7) ± 0.009 | 0.161(29) ± 0.009 | 0.153(23) ± 0.008 | - | **0.104(8)** ± 0.009 |
| Silhouettes | C | 0.39(14) ± 0.032 | 0.35(20) ± 0.024 | 0.378(16) ± 0.027 | 0.419(24) ± 0.03 | 0.297(17) ± 0.027 | 0.437(29) ± 0.022 | 0.389(28) ± 0.029 | 0.315(16) ± 0.042 | **0.248(8)** ± 0.029 |
| Raisin | C | 0.211(18) ± 0.02 | 0.195(26) ± 0.022 | 0.216(5) ± 0.021 | 0.216(3) ± 0.019 | 0.204(4) ± 0.021 | 0.26(30) ± 0.021 | 0.22(12) ± 0.022 | 0.18(3) ± 0.018 | **0.176(19)** ± 0.019 |
| Concrete | R | 0.077(19) ± 0.005 | 0.066(6) ± 0.005 | 0.064(7) ± 0.005 | 0.08(30) ± 0.006 | 0.051(8) ± 0.005 | 0.085(29) ± 0.007 | 0.07(26) ± 0.005 | - | **0.036(26)** ± 0.006 |

### E.3 COMPARISON BETWEEN SDM AND SSDM

We present a comparison between SDM and SSDM, acknowledging that SDM, in principle, can be applied in semi-supervised settings, particularly when runtime is not a constraint. To address this, we have included a detailed comparison in Table 5, showcasing results for both SDM and SSDM across all datasets where only SDM results were previously reported. As highlighted in the table, SDM generally achieves slightly better performance than SSDM, consistent with our theoretical analysis. However, SSDM demonstrates superior performance in certain cases, such as the Yacht dataset and the toy problem introduced in this paper, as illustrated in Figure 3.

The reason why SSDM can surpass SDM in certain cases becomes evident when examining the components of the toy problem shown in Figures 11 and 12. Figure 11 presents the SDM results with the optimal $t = 0.02$, while Figure 12 displays the SSDM results with the optimal $t = 0.8$. From these figures, we observe that SDM embeddings, influenced by individual kernels for each data point, exhibit greater variability, leading to embeddings that are less consistent across samples. In contrast, SSDM embeddings, constructed from a single kernel, demonstrate greater consistency across samples.

Theoretically, this variability in SDM could be mitigated by using larger kernels. However, SDM becomes impractical for large datasets due to its computational complexity.

In conclusion, while SSDM is significantly more time-efficient and its performance is comparable to SDM, SDM remains relevant for fully supervised settings and can achieve better performance in some cases.

Table 5: Comparison between SDM and SSDM, reporting NMSE for regression ('R') and Misclassification Rate for classification ('C'), along with runtimes (in seconds).

| Dataset | | SDM | | SSDM | |
|---|---|---|---|---|---|
| Name | Type | Error | Runtime | Error | Runtime |
| Iris | C | **0.034 ± 0.026** | 5.5 | 0.036 ± 0.022 | 0.1 |
| Ionosphere | C | **0.073 ± 0.023** | 50 | 0.098 ± 0.023 | 0.4 |
| Arrhythmia | C | **0.431 ± 0.048** | 107 | **0.431 ± 0.038** | 0.7 |
| Musk | C | **0.101 ± 0.024** | 124 | 0.124 ± 0.023 | 0.5 |
| Yacht | R | 0.12 ± 0.063 | 9.3 | **0.008 ± 0.003** | 0.1 |
| Boston | R | 0.068 ± 0.01 | 41 | **0.05 ± 0.008** | 0.3 |
| Liver | R | **0.474 ± 0.054** | 13 | 0.48 ± 0.051 | 0.1 |

### E.4 ERROR RATES ACROSS DIMENSIONALITY

SDM achieves the best results for all datasets, and in some cases, its performance is superior across all chosen dimensions. For instance, as shown in Figure 15, SDM outperforms all evaluated algorithms for the Ionosphere dataset across all dimensions from 1 to 30.

Moreover, SSDM achieves the best result for all datasets except for Mice, and in some cases, its performance is superior across most chosen dimensions. For instance, as shown in Figure 16, our SSDM outperforms all evaluated algorithms for the Silhouettes dataset across all dimensions from 3 to 30.

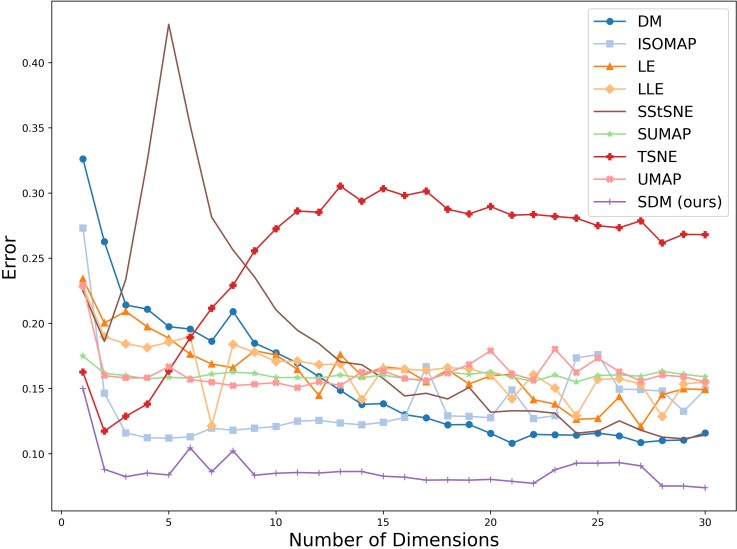

Figure 15: This figure depicts the error rate for the Ionosphere dataset as the number of dimensions increases in the supervised setting.

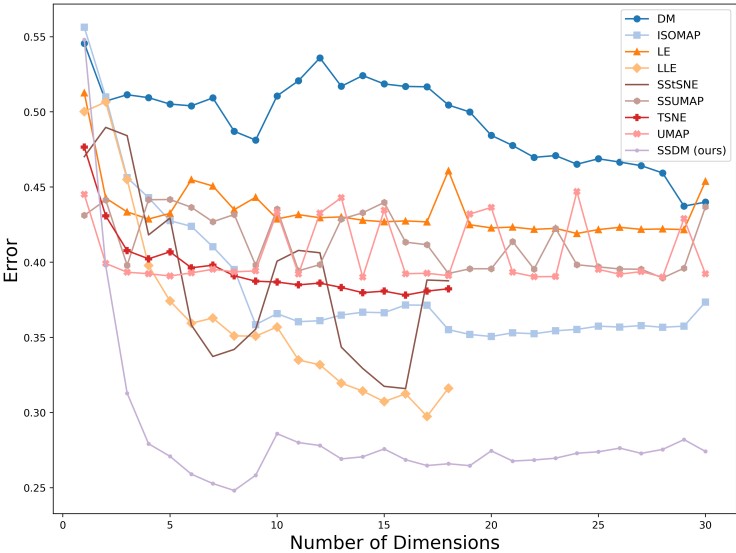

Figure 16: This figure depicts the error rate for the Silhouettes dataset as the number of dimensions increases in the semi-supervised setting.

## E.5 RUNTIME COMPARISON

Table 6: Comparison of Runtimes [in secs]

| Dataset | UMAP | Isomap | tSNE | LE | LLE | DM | SUMAP | SSUMAP | SStSNE | SDM | SSDM |
|---------|------|--------|------|------|------|------|-------|--------|--------|-----|------|
| Iris | 2.5 | 0.01 | 0.5 | 0.01 | 0.01 | 0.02 | 4.3 | 2.6 | 0.8 | 5.5 | 0.1 |
| Mice | 3 | 0.5 | 47 | 0.5 | 0.2 | 4.1 | 5 | 3.5 | 29.4 | 934 | 3 |
| Rice | 7 | 2.7 | 110 | 0.7 | 1 | 16.2 | 8.5 | 7.8 | N/A | N/A | 55 |

## F    COMPLEXITY ANALYSIS AND OPTIMIZED SSDM IMPLEMENTATION

For SSDM, the affinity kernels $\mathbf{D}$ and $\mathbf{P}$ are computed with a time complexity of $O(d \cdot n^2)$, where $d$ is the dimensionality of the data. The computation of $\mathbf{D}^t$ and $\mathbf{P}^{1-t}$ using SVD requires $O(n^3)$, and multiplying the kernels to obtain $\mathbf{P}^{1-t}\mathbf{D}^t$ also takes $O(n^3)$. Finally, obtaining the embeddings using SVD is an additional $O(n^3)$. Therefore, the overall time complexity of SSDM is $O(n^3)$, with a space complexity of $O(n^2)$.

In contrast, SDM involves applying the SSDM procedure $n$ times, resulting in a time complexity of $O(n^4)$ and a space complexity of $O(n^2)$. This analysis indicates that SDM is impractical for large or even medium-sized datasets, while SSDM remains feasible for datasets with up to approximately 10,000 samples.

To overcome this limitation, we have implemented an optimized version of SSDM that is highly suitable for large-scale datasets. In the optimized version, instead of constructing the label and data kernels $\mathbf{P}$ and $\mathbf{D}$ with dimensions $n \times n$, we randomly sample $k$ labeled samples from the training set. This results in a label kernel $\mathbf{P}$ of size $k \times k$ and a data kernel $\mathbf{D}$ of size $k \times n$, where $k$ is set in our experiments to $0.01 \cdot n$ if $n > 10,000$ and to $0.1 \cdot n$ if $n \leq 10,000$. Empirically, we have found that a small proportion of labeled samples can effectively represent the space in large datasets. For very large datasets, $k$ can be set to any fixed small value, smaller than $0.01 \cdot n$. Apart from the adjustments to the kernel dimensions, the other steps of the optimized SSDM remain identical to those in SSDM. This modification reduces the complexity of the optimized SSDM to $O(k^2 \cdot n)$, making it scalable for very large datasets (larger than 100,000 samples).

Furthermore, the optimized SSDM is implemented using `torch`, leveraging GPU acceleration to perform matrix multiplications in parallel. This allows for significant speedups in computation, enabling the method to process large-scale datasets efficiently.

For comparison, t-SNE has a time complexity of $O(n^2)$, with optimized versions having a complexity of $O(n \cdot \log n)$, while UMAP has a time complexity of $O(d \cdot n^{1.14})$. As such, for smaller values of $k$, our method is more time-efficient than these alternatives, as observed in our empirical experiments.

To evaluate the effectiveness of the optimized SSDM and the improvements in runtime, we follow the procedure outlined in Section 6, which involves reducing the dimensionality, training a KNN classifier, and reporting the Misclassification Rate.

First, we compare the optimized SSDM to the unoptimized SSDM on the Rice dataset (3810 samples). As reported in Table 6, the unoptimized SSDM takes 55 seconds. By contrast, using the optimized SSDM with $k = 0.1 \cdot n = 381$, we achieve a runtime of just 0.03 seconds, representing a $\times 1833$ improvement in runtime. The Misclassification Rate for the optimized SSDM is $0.133 \pm 0.013$, which is slightly worse than the $0.104 \pm 0.009$ reported for the unoptimized SSDM. However, this trade-off is accompanied by a significant improvement in runtime.

In Figure 17, we present heatmap visualizations (as discussed in Appendix C.1) comparing the unoptimized SSDM, which utilizes all training labels, to the optimized SSDM, which uses only $k = 381$ labels.

Next, we demonstrate the effectiveness of the optimized SSDM on large datasets that are impractical to process using the unoptimized SSDM. Specifically, we evaluate the optimized SSDM on the MNIST dataset (LeCun et al., 1998), Fashion-MNIST (Xiao et al., 2017), Isolet (Cole & Fanty, 1991), Adult (Becker & Kohavi, 1996), Landsat (Srinivasan, 1993), and Nursery (Rajkovic, 1989). In these experiments, the parameter $k$ is set to $0.01 \cdot n$ for $n > 10,000$ and $0.1 \cdot n$ for $n \leq 10,000$.

Table 7 presents the Misclassification Rate for each dataset. For comparison, we include results for the unsupervised Diffusion Maps (by using the data kernel $\mathbf{D}$ employed in the optimized SSDM, which has dimensions of $k \times n$) and semi-supervised UMAP. As shown in the table, our SSDM achieves the best performance on 3 out of the 6 datasets.

Table 8 reports the runtime comparisons. Notably, our optimized SSDM is significantly faster than semi-supervised UMAP across all datasets.

Table 7: Evaluation on Large Datasets: Comparison of Diffusion Maps, SSUMAP, and the Optimized SSDM

| Dataset | | | Misclassification Rate | | |
|---|---|---|---|---|---|
| Name | n | d | DM | SSUMAP | SSDM (ours) |
| MNIST | 70,000 | 784 | $0.22 \pm 0.006$ | $\mathbf{0.046 \pm 0.001}$ | $0.1 \pm 0.003$ |
| Fashion-MNIST | 70,000 | 784 | $0.33 \pm 0.016$ | $\mathbf{0.21 \pm 0.006}$ | $0.25 \pm 0.004$ |
| Isolet | 7797 | 617 | $0.147 \pm 0.003$ | $0.178 \pm 0.005$ | $\mathbf{0.087 \pm 0.006}$ |
| Adult | 48842 | 14 | $\mathbf{0.211 \pm 0.002}$ | $0.212 \pm 0.001$ | $0.23 \pm 0.003$ |
| Landsat | 6435 | 36 | $0.181 \pm 0.013$ | $\mathbf{0.127 \pm 0.004}$ | $\mathbf{0.127 \pm 0.007}$ |
| Nursery | 12960 | 8 | $0.128 \pm 0.03$ | $0.193 \pm 0.005$ | $\mathbf{0.082 \pm 0.008}$ |

Table 8: Runtimes on Large Datasets (in seconds): Comparison of SSUMAP and the Optimized SSDM

| Dataset | | | Runtime | |
|---|---|---|---|---|
| Name | n | d | SSUMAP | SSDM (ours) |
| MNIST | 70,000 | 784 | 35 | **8.1** |
| Fashion-MNIST | 70,000 | 784 | 35 | **8.35** |
| Isolet | 7797 | 617 | 7 | **1.2** |
| Adult | 48842 | 14 | 27 | **1.4** |
| Landsat | 6435 | 36 | 6.3 | **0.12** |
| Nursery | 12960 | 8 | 5.5 | **0.4** |

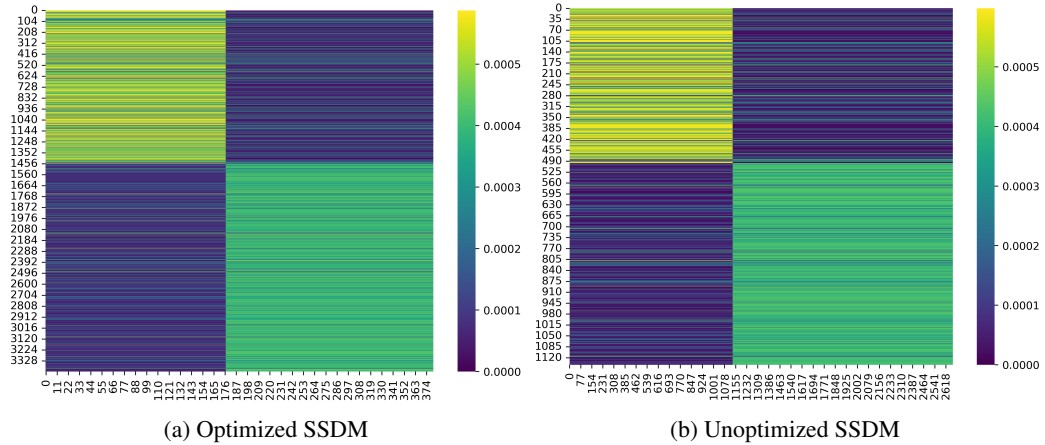

(a) Optimized SSDM

(b) Unoptimized SSDM

Figure 17: Heatmap visualization of the optimized SSDM kernel using only $k = 381$ labeled samples, compared to the unoptimized SSDM kernel utilizing all labeled samples for the Rice dataset.

## G    EXTENDED REVIEW AND FUTURE DIRECTIONS

### G.1    MULTI-VIEW LEARNING

Multi-view learning utilizes information from multiple representations or perspectives of the same data to enhance learning performance. These views can originate from different modalities, such as text, images, or audio, or from different transformations of the same modality (Xu et al., 2013). Traditional approaches like co-training (Kumar & Daumé, 2011) focus on alternating the training of classifiers on distinct views, leveraging the mutual information between them to improve performance. Canonical Correlation Analysis (CCA) is a foundational method that identifies linear transformations of two views to maximize their correlation in a shared subspace (Hotelling, 1936). Recent developments in multi-view deep learning have integrated these principles into neural networks, enabling models like multi-view autoencoders (Du et al., 2021) to jointly learn from heterogeneous data representations.

Recent advancements in multi-view learning include the emergence of co-regularization and margin-consistency algorithms (Zhao et al., 2017). Co-regularization approaches introduce constraints to ensure the consistency of predictions across views, utilizing methods like multi-view Support Vector Machines (SVMs) (Li et al., 2004) and multi-view linear discriminant analysis (Kan et al., 2015) to align feature spaces. Margin-consistency algorithms, on the other hand, enforce agreement on decision boundaries across views, often leveraging maximum entropy discrimination frameworks (Zhao et al., 2017). These methods demonstrate the power of multi-view learning in diverse tasks such as clustering, classification, and transfer learning, where the complementary information from multiple views significantly enhances performance.

A key challenge in multi-view learning is effectively integrating diverse and sometimes incomplete views in a scalable manner. Large-scale applications, such as analyzing multi-modal datasets (e.g., combining video, audio, and text), demand computationally efficient algorithms. Approaches like multi-view dimensionality reduction aim to create low-dimensional representations that preserve the shared structure among views while accounting for unique characteristics.

One branch of multi-view learning is multi-kernel learning (Bach et al., 2004), which combines multiple kernels, each capturing a specific view or aspect of the data, into a unified framework. By integrating diverse sources of information, kernel methods provide a flexible and powerful approach to learning relationships in complex datasets. They are particularly well-suited for multi-view settings because they can model nonlinear relationships and accommodate varying feature spaces between views. Kernel-based methods often employ techniques like kernel alignment or optimization of weights across kernels to balance the contributions of different views.

Alternating diffusion (Lederman & Talmon, 2018) is a prominent method within the multi-kernel learning framework. Our work incorporates elements of multi-view learning by treating labels as a second view of the data. While we focus on alternating diffusion to integrate label information, similar strategies can be applied in other multi-view methods to incorporate labels and enhance the learning process.

### G.2    EXTENDED REVIEW OF ALTERNATING DIFFUSION

Alternating Diffusion (AD) (Lederman & Talmon, 2018) extends the Diffusion Maps framework, and diffusion geometry (Coifman & Lafon, 2006; Lin et al., 2024), to extract the common structure of two aligned datasets,

$$\{x_i^{(1)}\}_{i=1}^n \quad , \quad \{x_i^{(2)}\}_{i=1}^n.$$

This technique is particularly useful for analyzing multi-view data where the goal is to disentangle shared latent structures while suppressing view-specific variability.

The process begins by constructing affinity matrices $\mathbf{W}^{(1)}$ and $\mathbf{W}^{(2)}$ for the datasets $\{x_i^{(1)}\}_{i=1}^n$ and $\{x_i^{(2)}\}_{i=1}^n$, respectively. The entries of these matrices are computed using a Gaussian kernel:

$$\mathbf{W}^{(v)}(i,j) = \exp\left(-\frac{d(x_i^{(v)}, x_j^{(v)})^2}{\epsilon^{(v)}}\right), \quad v \in \{1, 2\},$$

where $d(\cdot, \cdot)$ is a distance metric, and $\epsilon^{(v)}$ is the kernel scale parameter for view $v$. These affinity matrices encode local pairwise similarities within each dataset.

To construct diffusion operators, the affinity matrices are normalized in two stages. First, a diagonal matrix $\mathbf{D}_1^{(v)}$ is computed from the row sums of $\mathbf{W}^{(v)}$:

$$\mathbf{D}_1^{(v)}(i, i) = \sum_j \mathbf{W}^{(v)}(i, j).$$

The affinity matrix is then symmetrically normalized to obtain:

$$\widetilde{\mathbf{K}}^{(v)} = (\mathbf{D}_1^{(v)})^{-1} \mathbf{W}^{(v)} (\mathbf{D}_1^{(v)})^{-1}.$$

Next, a second diagonal matrix $\mathbf{D}_2^{(v)}$ is calculated from the row sums of $\widetilde{\mathbf{K}}^{(v)}$:

$$\mathbf{D}_2^{(v)}(i, i) = \sum_j \widetilde{\mathbf{K}}^{(v)}(i, j).$$

Finally, the row-stochastic diffusion operator is obtained as:

$$\mathbf{K}^{(v)} = (\mathbf{D}_2^{(v)})^{-1} \widetilde{\mathbf{K}}^{(v)}.$$

This matrix, $\mathbf{K}^{(v)}$, represents the transition probability matrix of a Markov chain on the dataset for view $v$.

The alternating diffusion process combines the diffusion operators from the two views. The combined operator, $\mathbf{K}^{(1) \cap (2)}$, is defined as:

$$\mathbf{K}^{(1) \cap (2)} = \mathbf{K}^{(1)} \mathbf{K}^{(2)}.$$

This operator alternates between propagating information through $\mathbf{K}^{(1)}$ and $\mathbf{K}^{(2)}$, facilitating the extraction of shared structures between the datasets.

To extract the shared latent structure, spectral decomposition is performed on $\mathbf{K}^{(1) \cap (2)}$. The eigenvectors corresponding to the largest eigenvalues provide the embedding coordinates:

$$\mathbf{K}^{(1) \cap (2)} \mathbf{v}_j = \mu_j \mathbf{v}_j, \quad j = 1, 2, \ldots, n,$$

where $\mathbf{v}_j$ are the embedding vectors, and $\mu_j$ are the eigenvalues. The dominant eigenvectors represent the smoothest variations that are consistent across both datasets.

In conclusion, Alternating Diffusion extends the Diffusion Maps framework to capture the common structure between two aligned datasets by using an alternating diffusion process that propagates information back and forth across the two modalities. This process can be interpreted as diffusion on two distinct graphs, one for each dataset. In each diffusion step, the kernel corresponds to a transition matrix on the respective graph, where mass is propagated across the vertices (samples) based on the kernel's transition probabilities. As the process alternates between the two kernels, the mass spreads across both graphs, ensuring that information from both datasets influences each other. This alternating propagation gradually aligns shared structures while suppressing dataset-specific variations, resulting in a smooth representation that highlights the common features between the datasets and effectively extracts the shared latent structure while filtering out modality-specific noise.

### G.3 FUTURE WORK

One possible direction for future work is to explore other kernel interpolation schemes for integrating data and label information. While our current method uses a specific approach to combine the kernels, alternative schemes may provide better performance or adapt more effectively to different datasets. Investigating these alternatives could enhance the flexibility and robustness of the framework.

Another promising idea is to learn the amplitude of each spectral component (eigenvalue) instead of relying on the power mechanism we currently use, where the eigenvalues are raised to $t$ and $1 - t$. Our current approach maintains the relative order of the components for any value of $t$, with the leading eigenvalues of $\mathbf{P}$ and $\mathbf{D}$ always remaining dominant. By explicitly learning the eigenvalues,

it would be possible to adjust the order and influence of components dynamically, potentially leading to improved performance by tailoring the spectral properties to specific tasks or datasets.

Additionally, extending our multi-view approach to other algorithms, as we have done here with alternating diffusion, is another avenue worth exploring. By applying the same principles to methods like multi-view autoencoders, we could enable these approaches to better leverage label information and enhance their effectiveness in classification, regression, or clustering tasks.

Lastly, as we have demonstrated the effectiveness of our label-driven diffusion in a graph setting, we believe it could also be beneficial to explore its incorporation in the context of geometric deep learning. Geometric deep learning methods, which work on non-Euclidean data such as graphs and manifolds, could further benefit from our approach by capturing complex relationships between data and labels in a way that respects the underlying geometry of the data.

