# OpenReview forum: "Supervised and Semi-Supervised Diffusion Maps with Label-Driven Diffusion"
_ICLR.cc/2025/Conference — ICLR 2025 Poster_

### Official Review · Reviewer_oUg2 · 2024-10-21

**Soundness:** 2
**Presentation:** 2
**Contribution:** 2
**Rating:** 6
**Confidence:** 2

**Summary:**

This paper proposes constructing two affinity kernels corresponding to the data and the labels. The authors then propose a multiplicative interpolation scheme of the two kernels, whose purpose is twofold. First, their scheme extracts the common structure underlying the data and the labels by defining a diffusion process driven by the data and the labels. This label-driven diffusion produces an embedding that emphasizes the properties relevant to the label-related task. Second, the proposed interpolation scheme balances the influence of the two kernels. We show on multiple benchmark datasets that the embedding learned by SDM and SSDM is more effective in downstream regression and classification tasks than existing unsupervised, supervised, and semi-supervised nonlinear dimension reduction methods.

**Strengths:**

1. Originality. This paper presents Supervised Diffusion Maps (SDM) and Semi-Supervised Diffusion Maps (SSDM), which are supervised and semi-supervised variants of Diffusion Maps. Unlike existing supervised and semi-supervised manifold learning methods,
the authors present a new approach, where they view the labels as an additional data modality and employ concepts of multimodal manifold learning. Concretely, the authors propose to construct a separate affinity kernel for the labels, allowing us to capture the underlying geometry of the labels in addition to that of the data.
2. Quality and Clarity. There are no technical errors, and the presentation and writing are satisfied.
3. Significance. The authors show on multiple benchmark datasets that the embedding learned by SDM and SSDM is more effective in downstream regression and classification tasks than existing unsupervised, supervised, and semi-supervised nonlinear dimension reduction methods.

**Weaknesses:**

I am absolutely not in this field and the comments from me are not relatively professional. The comments in the following are just raised from the presentation or organization.

1. The authors are expected to give the figure of the framework based on Supervised Diffusion Maps (SDM) and Semi-Supervised Diffusion Maps (SSDM) and highlight the major differences between the proposed method and the most related work.

2. The authors use a new subsection termed alternating diffusion in Background part. However, there are just small amount of lines in this subsection. I think the authors can further improve this aspect for this paper.

3. For the evaluation results, the authors report NMSE for regression and Misclassification Rate for classification based on S/SDM for supervised SDM and semi-supervised SSDM on the given datasets. I think the authors can add more datasets for comparison in the experiment.

4. In the conclusion part, the authors explore the futher work by exploring the utility and adaptation of fusion of data and label kenels. The authors can add a new subsection to discuss the futher work in details here.

**Questions:**

1. Why the authors reduce the original data dimensionality to a range of 1 to 30 dimensions for each dataset and algorithm in the experiment?

2. Why the authors choose Euclidean distance in the data kernel and angular distance in the label kernel?

**Details Of Ethics Concerns:**

I have no details of Ethics Concerns for this work.

---

> ### Author Response · Authors · 2024-11-20
> **Response to reviewer oUg2 - part 1**
>
> Thanks for your constructive feedback. We appreciate your time and the helpful comments you've provided.
>
> **1. Framework figures and comparison with related work:** Thank you for your comment. In response to your feedback, we have added Appendix A.2 to the revised paper, which includes preliminary block diagram figures for both SDM and SSDM. We plan to enhance these diagrams for the camera-ready version, including improvements such as adding an illustration to show how $\mathbf{D}$ is constructed from the samples and $\mathbf{P}$ from the available labels.
>
> Regarding comparison with related work, section 2 (Related Work) discusses common approaches for incorporating supervision (labels) into manifold learning techniques. The key distinction of our method lies in our new approach of integrating supervision by treating the labels as a second modality. Furthermore, our approach builds on the unsupervised Diffusion Maps framework and introduces a label-driven diffusion process. This process extracts the shared geometry between the data and the labels, which significantly enhances the resulting embedding as we show on multiple real-world datasets in Section 6, especially for downstream tasks such as classification and regression.
>
> We have made changes in the revised version of the paper to more clearly highlight these differences, as you suggested. Additionally, we have added a new Appendix G, in which we extend the Related Work section and provide additional information about alternating diffusion. Furthermore, in the new Appendix C.1 of the revised version, we compare the performance of our label-driven diffusion against standard diffusion (used in the unsupervised Diffusion Maps) using new heatmap visualizations of the kernels. These results highlight the effectiveness of our approach.
>
>
> **2. Elaborating on alternating diffusion:** Thank you for your suggestion. In response, we have added a new Appendix G in the revised version of the paper, where we extend the Related Work section and provide a more detailed explanation of the alternating diffusion process, and . We hope this additional content will help to better clarify and expand on this important aspect of our method.
>
>
> **3. Evaluation on additional datasets:** Thank you for your suggestion. In the revised version, we have included additional evaluations of SSDM on the datasets previously reported for SDM. The results are now presented in Table 5 of the new Appendix E.3. Furthermore, in the new Appendix F, we evaluate 6 more large datasets, providing both the evaluation results and runtime comparisons. We kindly refer you to a separate post in this thread, where we present these results.
>
>
> **4. Discussion of future work:** Thank you for your suggestion and valuable input. As suggested, in the revised version of the paper, we have added more details about possible future work in Appendix G.3.
>
> One promising direction involves incorporating our multi-view approach for integrating labels into manifold learning methods, and applying it to other multi-view algorithms as we have done here with alternating diffusion.
>
> Additionally, we aim to explore other kernel interpolation schemes for combining data and label information. While the current approach works effectively, other interpolation methods could offer more flexibility and improve performance in different settings.
>
> Another potential direction is to move beyond the power mechanism we use for adjusting the influence of spectral components (eigenvalues). Instead of raising the eigenvalues to powers $t$ and $1-t$, we could learn the amplitude of each eigenvalue explicitly. This would allow us to modify the order and influence of spectral components dynamically, potentially leading to better performance by adjusting the contributions of each component to the learned embedding.
>
> Furthermore, as we have demonstrated the effectiveness of our label-driven diffusion in a graph setting, we believe it could also be beneficial to explore its incorporation in the context of geometric deep learning.

---

> ### Author Response · Authors · 2024-11-20
> **Response to reviewer oUg2 - part 2**
>
> ### Questions
> **1. Dimensionality reduction range:** Thank you for your question. You are correct that the number of reduced dimensions could be treated as a hyperparameter, and we could select the best number for each algorithm. However, we chose to reduce the dimensionality to a range of 1 to 30 dimensions to demonstrate that our method is insensitive to the choice of this parameter and performs well not only for the optimal number of dimensions but also across a broader range of dimensions. This helps highlight the robustness of our method. We have further elaborated on this aspect in Appendix E.4, where we show the performance of our method across different dimensionalities.
>
>
> **2. Choice of distance metrics for data and label kernels:** Thank you for your question. The choice of distance metric should ideally be based on the nature of the data and labels. For example, in the toy problem presented in Section 6, where the labels represent rotation angles, we used angular distance due to the cyclic nature of angles. We also observed that the Euclidean distance, although slightly worse, still produced reasonable results. For the real-world datasets evaluated in Section 6, we used the Euclidean distance for both the data and label kernels for simplicity. However, we acknowledge that carefully selecting more suitable distance metrics for each dataset could further improve the results.

---

> > ### Comment · Reviewer_oUg2 · 2024-11-26
> >
> > I have read this response and keep this positive score.

---

### Official Review · Reviewer_HxoS · 2024-11-01

**Soundness:** 3
**Presentation:** 2
**Contribution:** 2
**Rating:** 6
**Confidence:** 3

**Summary:**

The paper proposes Supervised Diffusion Maps (SDM) and Semi-Supervised Diffusion Maps (SSDM) as extensions of the classical Diffusion Maps algorithm. By treating labels as a secondary view of the data, the authors employ a multi-view learning framework to enhance the dimensionality reduction process. The proposed methods introduce a multiplicative kernel interpolation scheme to integrate data and label information. Experimental results on benchmark datasets are used to demonstrate the effectiveness of SDM and SSDM compared to existing manifold learning techniques.

**Strengths:**

1) The paper presents a clear motivation for addressing the limitations of traditional manifold learning methods that lack mechanisms to incorporate label information.
2) Treating labels as an additional data modality and leveraging a multi-view framework can provide some new insights to the community.
3) Both empirical and theoretical analyses are provided to justify the proposed method.

**Weaknesses:**

1. While the multi-view framework and the label-driven diffusion are interesting, they rely heavily on existing concepts, such as Alternating Diffusion and classical kernel methods. Thus, the novelty of the proposed SDM and SSDM methods appears limited, as they mainly extend well-established techniques.
2. Given that the paper relies heavily on older references with minimal discussion of more contemporary work, are there recent advancements in manifold learning？
3. There is no discussion on how the proposed methods scale with larger datasets (e.g. MNIST [1] with 70000 samples) or in higher-dimensional spaces (Toxicity [2] with 1203 features), which is crucial for practical applications.
4. The related work section lacks a comprehensive overview in multi-view learning [3,4].
5. The computational complexity of the proposed methods is not well-handled, with larger computational complexity compared with other methods.

[1] LeCun Y, Bottou L, Bengio Y, et al. Gradient-based learning applied to document recognition[J]. Proceedings of the IEEE, 1998, 86(11): 2278-2324.
[2] Gul S, Rahim F, Isin S, et al. Structure-based design and classifications of small molecules regulating the circadian rhythm period[J]. Scientific reports, 2021, 11(1): 18510.
[3] Lyu G, Yang Z, Deng X, et al. L-VSM: Label-Driven View-Specific Fusion for Multiview Multilabel Classification[J]. IEEE Transactions on Neural Networks and Learning Systems, 2024.
[4] Zhao J, Xie X, Xu X, et al. Multi-view learning overview: Recent progress and new challenges[J]. Information Fusion, 2017, 38: 43-54.

**Questions:**

Please refer to the weakness list.

---

> ### Author Response · Authors · 2024-11-20
> **Response to reviewer HxoS - part 1**
>
> Thank you for your time and feedback. We appreciate your suggestions and the opportunity to improve our paper.
>
> **1. The novelty in our method:** We appreciate your feedback, but we believe there may be a misunderstanding regarding the novelty of our approach. While it is true that we build on alternating diffusion and kernel methods, the key innovation in our work lies in how we incorporate labels as a distinct data modality within a multi-view learning framework. To the best of our knowledge, this is the first work to introduce this novel perspective, where labels are treated as a second data modality and integrated into the dimension reduction process to guide learning. This approach has the potential to be applied to other multi-view learning algorithms as well.
>
> Furthermore, we extend alternating diffusion to a partial alignment setting to address the challenge of handling unlabeled data, which the original alternating diffusion framework does not accommodate. Our solution, from a graph perspective, involves adding isolated nodes to the label graph to handle unlabeled samples that lack direct alignment with labels, a crucial aspect for supervised and semi-supervised settings. We also provide a theoretical justification for this extension to the partial alignment setting.
>
> In the new Appendix C.1 in the revised version of the paper, we compare the performance of our label-driven diffusion against standard diffusion using heatmap visualizations of the kernels. These results highlight the effectiveness of our approach.
>
>
> **2. Comparison with recent manifold learning methods:** Thank you for your comment. We believe we have addressed the most relevant work in the field. In our evaluation, we compare our method to UMAP [1], which is often considered one of the gold standards today for dimensionality reduction, along with other classical methods.
>
> While a few newer methods such as TriMap [2], PaCMAP [3], and TopoMap [4] have garnered attention within the community, they do not currently support label incorporation, which makes them less suitable for supervised or semi-supervised settings like ours. Furthermore, while these newer approaches bring interesting ideas to the table with their own advantages and limitations, none have yet been conclusively proven to outperform UMAP or other established algorithms across all scenarios.
>
> For these reasons, we chose to focus our comparisons on well-known algorithms that have demonstrated broad effectiveness and include established supervised and semi-supervised variants, which are more aligned with the goals of our work. Comparing our approach to methods like PaCMAP, which lacks a clear advantage over other algorithms, would not provide additional insights beyond the comparison with UMAP.
>
> This practice of comparing to the most prominent methods in recent years is common within the manifold learning community. For example, the recently published manifold learning algorithm HeatGeo [5], presented at NeurIPS 2024, includes comparisons to UMAP [1], t-SNE [6], Isomap [7], Diffusion Maps [8], and PHATE [9]. In our work, we compare against all of them, except for PHATE. Additionally, we compare against other methods such as Laplacian Eigenmaps, Locally Linear Embedding [10], supervised and semi-supervised UMAP, and semi-supervised t-SNE.
>
> [1]McInnes, Leland, John Healy, and James Melville. "UMAP: Uniform manifold approximation and projection for dimension reduction." arXiv preprint arXiv:1802.03426 (2018).
>
> [2]Amid, Ehsan, and Manfred K. Warmuth. "TriMap: Large-scale dimensionality reduction using triplets." arXiv preprint arXiv:1910.00204 (2019).
>
> [3]Wang, Yingfan, et al. "Understanding how dimension reduction tools work: an empirical approach to deciphering t-SNE, UMAP, TriMAP, and PaCMAP for data visualization." Journal of Machine Learning Research 22.201 (2021): 1-73.
>
> [4]Doraiswamy, Harish, et al. "TopoMap: A 0-dimensional homology preserving projection of high-dimensional data." IEEE Transactions on Visualization and Computer Graphics 27.2 (2020): 561-571.
>
> [5]Huguet, Guillaume, et al. "A heat diffusion perspective on geodesic preserving dimensionality reduction." Advances in Neural Information Processing Systems 36 (2024).
>
> [6]Van der Maaten, Laurens, and Geoffrey Hinton. "Visualizing data using t-SNE." Journal of machine learning research 9.11 (2008).
>
> [7]Balasubramanian, Mukund, and Eric L. Schwartz. "The isomap algorithm and topological stability." Science 295.5552 (2002): 7-7.
>
> [8]Coifman, Ronald R., and Stéphane Lafon. "Diffusion maps." Applied and computational harmonic analysis 21.1 (2006): 5-30.
>
> [9]Moon, Kevin R., et al. "PHATE: a dimensionality reduction method for visualizing trajectory structures in high-dimensional biological data." BioRxiv 120378 (2017).
>
> [10] Roweis, Sam T., and Lawrence K. Saul. "Nonlinear dimensionality reduction by locally linear embedding." science 290.5500 (2000): 2323-2326.

---

> ### Author Response · Authors · 2024-11-20
> **Response to reviewer HxoS - part 2**
>
> **3. Scaling up:** We appreciate your comment regarding our method scale to larger datasets and higher-dimensional spaces, and we have addressed these concerns in the revised version of our paper.
>
> In response, we have added a new Appendix F where we provide a detailed analysis of the time and space complexity of both SDM and SSDM. The complexity of SDM is $O(n^4)$, which makes it impractical for large or even medium-sized datasets. Conversely, SSDM has a complexity of $O(n^3)$, making it more feasible for medium-sized datasets, up to approximately 10,000 samples.
>
> In general, large datasets pose a known challenge in kernel methods like ours. Typically, using sparse kernels reduces both computational and storage demands. However, since our method involves kernel matrix multiplication and since the multiplication of two sparse matrices may result in a dense matrix, this approach is not suitable for scaling up our method. One possible solution is the "landmark" [1] approach, which has been adapted for alternating diffusion in recently [2]. This approach approximates the eigenvectors of large alternating diffusion kernels by sampling $k$ landmark data points, computing the eigenvectors for the small $k \times k$ kernel, and then expanding them to the entire dataset.
>
> To address this limitation, we have implemented a simpler sampling approach to create an optimized version of SSDM that is suitable for large-scale datasets. For example, on our basic hardware (described in Appendix B), the optimized SSDM processes datasets like MNIST and Fashion-MNIST (each containing 70,000 samples) in just 8 seconds. Additionally, the Rice dataset, which contains 3,810 samples and took 55 seconds to process with SSDM (as reported in Table 6), is now processed in just 0.03 seconds using the optimized SSDM. The optimized SSDM efficiently handles high-dimensional data, as demonstrated by its performance on MNIST (784 features), Fashion-MNIST (784 features), and Isolet (617 features).
>
> In the optimized version, rather than constructing the label and data kernels $\mathbf{P}$ and $\mathbf{D}$ with dimensions $n \times n$, we randomly sample $k$ labeled samples from the training set. This results in a label kernel $\mathbf{P}$ of size $k \times k$ and a data kernel $\mathbf{D}$ of size $k \times n$, where $k$ is set to $0.01 \cdot n$ if $n > 10,000$ and to $0.1 \cdot n$ if $n \leq 10,000$ in our experiments. Through empirical testing, we found that a small proportion of labeled samples can effectively represent the space in large datasets. If $n$ becomes very large, $k$ can be set to any fixed small value, smaller than $0.01 \cdot n$. Apart from adjusting the kernel dimensions, the other steps of the optimized SSDM remain identical to those in SSDM. This modification reduces the complexity of the optimized SSDM to $O(k^2 \cdot n)$, making it scalable for very large datasets (greater than 100,000 samples). Additionally, the optimized SSDM is implemented using torch, which leverages GPU acceleration.
>
> For comparison, t-SNE has a time complexity of $O(n^2)$, with optimized versions having a complexity of $O(n \cdot \log n)$, while UMAP has a time complexity of $O(d \cdot n^{1.14})$, where $d$ is the dimensionality of the data. As such, for smaller values of $k$, our method is more time-efficient than these alternatives.
>
> In the newly added Appendix F, we provide a comprehensive description of the optimized SSDM, along with detailed performance and runtime comparisons based on the procedure outlined in the Experimental Results section. Additionally, we wish to kindly refer you to our general post in this thread, where we present both evaluation results and runtime comparisons on large datasets, comparing between optimized SSDM, unsupervised Diffusion Maps, and semi-supervised UMAP. In which our SSDM achieves the best performance on 3 out of the 6 datasets, and is significantly faster than semi-supervised UMAP across all datasets. We will upload the code for the optimized SSDM in the supplementary materials within the next few days.
>
> [1] Shen, Chao, and Hau-Tieng Wu. "Scalability and robustness of spectral embedding: landmark diffusion is all you need." Information and Inference: A Journal of the IMA 11.4 (2022): 1527-1595.
>
> [2] Yeh, Sing-Yuan, et al. "Landmark Alternating Diffusion." arXiv preprint arXiv:2404.19649 (2024).
>
>
>
> **4. Related work:** Thank you for your comment on the related work section. In response to your feedback, we have added Appendix G in the revised version of the paper, which provides a comprehensive overview of key approaches in multi-view learning. Specifically, we discuss methods such as co-training, canonical correlation analysis (CCA), multi-kernel learning methods (e.g, alternating diffusion), and multi-view deep learning.

---

> ### Author Response · Authors · 2024-11-20
> **Response to reviewer HxoS - part 3**
>
> **5. Computational complexity:** We provide a detailed analysis of the time and space complexity of both SDM and SSDM in the new Appendix F. The complexity of SDM is $O(n^4)$, making it impractical for large or medium-sized datasets, while SSDM has a complexity of $O(n^3)$, suitable for datasets up to 10,000 samples.
>
> The optimized SSDM, also detailed in the new Appendix F, has a complexity of $O(k^2 \cdot n)$, making it scalable for very large datasets (over 100,000 samples). Additionally, it is implemented using torch to leverage GPU acceleration.
>
> For comparison, t-SNE has a complexity of $O(n^2)$ (optimized versions: $O(n \cdot \log n)$), and UMAP has a time complexity of $O(d \cdot n^{1.14})$, where $d$ is the data dimensionality. Our method, for smaller values of $k$, is more time-efficient than these alternatives.

---

> ### Comment · Reviewer_HxoS · 2024-11-22
>
> Thanks for the rebuttals, which have addressed my concerns well. I have also read other reviewer's comments. I am willing to increase my score from 5 to 6.

---

> > ### Author Response · Authors · 2024-11-23
> >
> > Thank you for your valuable feedback and decision to raise the score.
> > Just a friendly reminder to kindly update the score when possible, as we noticed it hasn’t been reflected yet.

---

### Official Review · Reviewer_fvWq · 2024-11-04

**Soundness:** 3
**Presentation:** 3
**Contribution:** 3
**Rating:** 8
**Confidence:** 3

**Summary:**

This paper introduces two novel methods for manifold learning: Supervised Diffusion Maps (SDM) and Semi-Supervised Diffusion Maps (SSDM), which extend the traditional Diffusion Maps framework to supervised and semi-supervised settings. Unlike prior methods, SDM and SSDM treat labels as a secondary data modality, employing principles from multimodal manifold learning to better capture the structure of labeled data. The authors propose constructing a separate affinity kernel for the labels, enabling the model to represent the geometry of labels along with the data. Using an Alternating Diffusion (AD) approach, they introduce a kernel interpolation scheme that combines data and label affinity kernels. This creates a “label-driven diffusion” process, approximating a continuous two-step diffusion on the manifold that emphasizes features relevant to the task.

The main contributions are as follows:

1. Multi-View Approach: SDM and SSDM incorporate labels as an additional information source, enhancing the manifold learning process.
2. New Kernel Interpolation Scheme: A method for combining affinity kernels to reveal shared structures between data and labels while controlling their respective contributions.
3. Experimental Validation: Results show that SDM and SSDM outperform existing nonlinear manifold learning methods across several benchmark datasets, highlighting their effectiveness.

**Strengths:**

This paper introduces Supervised Diffusion Maps (SDM) and Semi-Supervised Diffusion Maps (SSDM), extending Diffusion Maps to supervised and semi-supervised learning by treating labels as an additional data modality. This innovative approach uses a two-step diffusion process, first on the labels and then on the data, enabling a “label-driven diffusion” that highlights task-relevant structures. The paper presents a rigorous kernel interpolation scheme to merge data and label affinities, supported by strong theoretical justification and validated through experiments on benchmark datasets. The methodology is clear, original, and impactful, making this a valuable contribution to manifold learning.

**Weaknesses:**

Diffusion processes are often computationally intensive, especially on large datasets. The paper would benefit from an analysis of the scalability of the proposed methods. Discussing time complexity, memory requirements, and potential optimizations or parallelization strategies could make this approach more practical for large-scale applications. Furthermore, a comparison of computational costs between SDM/SSDM and other manifold learning methods would provide a clearer picture of their efficiency.

**Questions:**

see weaknesses

---

> ### Author Response · Authors · 2024-11-20
> **Response to reviewer fvWq**
>
> Thanks for your positive review and thoughtful comments. We appreciate your time and the effort you put into reviewing our paper.
>
> **Computational Complexity and Scale Up:** We appreciate your comment regarding the computational complexity and scale up, and we have addressed these concerns in the revised version of our paper.
>
> In response, we have added a new Appendix F where we provide a detailed analysis of the time and space complexity of both SDM and SSDM. The complexity of SDM is $O(n^4)$, which makes it impractical for large or even medium-sized datasets. Conversely, SSDM has a complexity of $O(n^3)$, making it more feasible for medium-sized datasets, up to approximately 10,000 samples.
>
> In general, large datasets pose a known challenge in kernel methods like ours. Typically, using sparse kernels reduces both computational and storage demands. However, since our method involves kernel matrix multiplication and since the multiplication of two sparse matrices may result in a dense matrix, this approach is not suitable for scaling up our method. One possible solution is the "landmark" [1] approach, which has been adapted for alternating diffusion in recently [2]. This approach approximates the eigenvectors of large alternating diffusion kernels by sampling $k$ landmark data points, computing the eigenvectors for the small $k \times k$ kernel, and then expanding them to the entire dataset.
>
> To address this limitation, we have implemented a simpler sampling approach to create an optimized version of SSDM that is suitable for large-scale datasets. For example, on our basic hardware (described in Appendix B), the optimized SSDM processes datasets like MNIST and Fashion-MNIST (each containing 70,000 samples) in just 8 seconds. Additionally, the Rice dataset, which contains 3,810 samples and took 55 seconds to process with SSDM (as reported in Table 6), is now processed in just 0.03 seconds using the optimized SSDM.
>
> In the optimized version, rather than constructing the label and data kernels $\mathbf{P}$ and $\mathbf{D}$ with dimensions $n \times n$, we randomly sample $k$ labeled samples from the training set. This results in a label kernel $\mathbf{P}$ of size $k \times k$ and a data kernel $\mathbf{D}$ of size $k \times n$, where $k$ is set to $0.01 \cdot n$ if $n > 10,000$ and to $0.1 \cdot n$ if $n \leq 10,000$ in our experiments. Through empirical testing, we found that a small proportion of labeled samples can effectively represent the space in large datasets. If $n$ becomes very large, $k$ can be set to any fixed small value, smaller than $0.01 \cdot n$. Apart from adjusting the kernel dimensions, the other steps of the optimized SSDM remain identical to those in SSDM. This modification reduces the complexity of the optimized SSDM to $O(k^2 \cdot n)$, making it scalable for very large datasets (greater than 100,000 samples). Additionally, the optimized SSDM is implemented using torch, which leverages GPU acceleration.
>
> For comparison, t-SNE has a time complexity of $O(n^2)$, with optimized versions having a complexity of $O(n \cdot \log n)$, while UMAP has a time complexity of $O(d \cdot n^{1.14})$, where $d$ is the dimensionality of the data. As such, for smaller values of $k$, our method is more time-efficient than these alternatives.
>
> In the newly added Appendix F, we provide a comprehensive description of the optimized SSDM, along with detailed performance and runtime comparisons based on the procedure outlined in the Experimental Results section. Additionally, we wish to kindly refer you to our general post in this thread, where we present both evaluation results and runtime comparisons on large datasets, comparing between optimized SSDM, unsupervised Diffusion Maps, and semi-supervised UMAP. In which our SSDM achieves the best performance on 3 out of the 6 datasets, and is significantly faster than semi-supervised UMAP across all datasets. We will upload the code for the optimized SSDM in the supplementary materials within the next few days.
>
> [1] Shen, Chao, and Hau-Tieng Wu. "Scalability and robustness of spectral embedding: landmark diffusion is all you need." Information and Inference: A Journal of the IMA 11.4 (2022): 1527-1595.
>
> [2] Yeh, Sing-Yuan, et al. "Landmark Alternating Diffusion." arXiv preprint arXiv:2404.19649 (2024).

---

> > ### Comment · Reviewer_fvWq · 2024-11-26
> >
> > Thanks for the rebuttals, which have addressed my concerns well. I will keep my score.

---

### Official Review · Reviewer_7iCi · 2024-11-04

**Soundness:** 3
**Presentation:** 3
**Contribution:** 2
**Rating:** 6
**Confidence:** 4

**Summary:**

This paper extends traditional diffusion maps by integrating label information to create supervised and semi-supervised versions, known as Supervised Diffusion Maps (SDM) and Semi-Supervised Diffusion Maps (SSDM). The core algorithm involves constructing two affinity kernels—one for the data and one for the labels. These affinity matrices are then combined using an alternating diffusion (AD) approach, which extracts the shared structure between data and labels, producing embeddings that incorporate label information. Theoretical analysis and experimental results on several real datasets demonstrate that the proposed methods outperform existing unsupervised and supervised algorithms in classification accuracy and regression error.

**Strengths:**

The strengths of the paper are as follows:

- Presentation: The paper is well-written and easy to follow, with clear introductions to the background, problem formulation, and related work. The proposed method is thoroughly described and demonstrated.

- Originality: The proposed method is both novel and conceptually sound. The problem is well-motivated by highlighting that unsupervised manifold learning does not fully utilize label information, often resulting in sub-optimal performance. Introducing the alternating diffusion approach with an additional parameter $t$ for fusing and balancing data and label information is an effective and innovative idea.

- Theory: The theoretical analysis, while based on several fundamental assumptions, is valuable and provides important insights.

**Weaknesses:**

My concern to this paper mainly falls onto the following three aspects:

- **Lack of Discussion on the Geometric Similarity Assumption**: A fundamental assumption in the proposed method is the strong geometric similarity between the data and labels. However, it is unclear how this similarity impacts the method's performance. For instance, if the labels are only weakly correlated with the data geometry, would the proposed method perform worse than traditional unsupervised algorithms? An empirical or theoretical study on this factor would be invaluable as a guide for real-world applications. Currently, no such study exists, and all experiments seem to assume this strong similarity.

- **Computational Complexity and Dataset Scale**: Given that the computational complexity increases with training data size, this could be a critical factor in determining the method’s applicability. A quantitative complexity analysis of the proposed method, and possibly other baseline methods, would be beneficial, in addition to the runtime reported in Table 5. Moreover, the real datasets in the paper are relatively small (<4k samples), which may not reflect the scale commonly encountered in real applications. It would be informative to evaluate both performance and runtime as dataset size grows, particularly examining the differences between SDM and SSDM under larger-scale conditions.

- **Performance Comparison between SDM and SSDM**: Given that computational complexity is crucial and that SSDM is more computationally efficient than SDM, a direct performance comparison between SDM and SSDM would be valuable. It’s important to understand if SDM consistently outperforms SSDM despite the higher complexity, or if there are cases where SSDM could exceed SDM’s performance—particularly when the geometry of the test data is beneficial in constructing the data affinity matrix.

Addressing these questions would greatly enhance the paper and make it a strong fit for this venue.

**Questions:**

See weakness above.

---

> ### Author Response · Authors · 2024-11-20
> **Response to reviewer 7iCi - part 1**
>
> Thank you for your helpful feedback. We appreciate the time you took to review our work and provide valuable suggestions.
>
> **1. Geometric Similarity Assumption:** Thank you for this insightful comment. We agree that the geometric similarity assumption is central, and your observation prompted us to elaborate further on its role in the paper.
>
> In response to your feedback, we have added Appendix C.1, where we discuss the geometric similarity assumption and propose an alternative assumption that is more practical to quantify, as the geometric similarity can be challenging to measure for a given dataset.
>
> Briefly, the alternative assumption can be described as follows: if two samples have similar labels (small label distance), our label-driven diffusion process should produce smaller distances and higher transition probabilities than the direct pairwise distances in the unsupervised Diffusion Maps. In cases where direct distances better represent label distances, unsupervised Diffusion Maps is likely to outperform our method.
>
> In Appendix C.1, we test this assumption on real-world datasets and demonstrate that when it holds, our method outperforms unsupervised Diffusion Maps. At the same time, when it does not hold, Diffusion Maps may perform better. To visually evaluate whether the assumption holds, we present the kernels as heatmaps and compare our method's kernel to the unsupervised Diffusion Maps kernel. These heatmaps, sorted by label, with rows for unlabeled samples and columns for labeled samples, highlight the effectiveness of our label-driven diffusion relative to the direct pairwise distances used in Diffusion Maps. Note that an ideal heatmap for classification displays high-values in diagonal blocks corresponding to the classes. We have demonstrated in the new Appendix C.1 that our method achieves better performance when our kernel's heatmap more closely resembles the ideal block diagonal matrix than the Diffusion Maps heatmap across several datasets.

---

> ### Author Response · Authors · 2024-11-20
> **Response to reviewer 7iCi - part 2**
>
> **2. Computational Complexity and Scale Up:** We appreciate your comment regarding the computational complexity and scale up, and we have addressed these concerns in the revised version of our paper.
>
> In response, we have added a new Appendix F where we provide a detailed analysis of the time and space complexity of both SDM and SSDM. The complexity of SDM is $O(n^4)$, which makes it impractical for large or even medium-sized datasets. Conversely, SSDM has a complexity of $O(n^3)$, making it more feasible for medium-sized datasets, up to approximately 10,000 samples.
>
> In general, large datasets pose a known challenge in kernel methods like ours. Typically, using sparse kernels reduces both computational and storage demands. However, since our method involves kernel matrix multiplication and since the multiplication of two sparse matrices may result in a dense matrix, this approach is not suitable for scaling up our method. One possible solution is the "landmark" [1] approach, which has been adapted for alternating diffusion in recently [2]. This approach approximates the eigenvectors of large alternating diffusion kernels by sampling $k$ landmark data points, computing the eigenvectors for the small $k \times k$ kernel, and then expanding them to the entire dataset.
>
> To address this limitation, we have implemented a simpler sampling approach to create an optimized version of SSDM that is suitable for large-scale datasets. For example, on our basic hardware (described in Appendix B), the optimized SSDM processes datasets like MNIST and Fashion-MNIST (each containing 70,000 samples) in just 8 seconds. Additionally, the Rice dataset, which contains 3,810 samples and took 55 seconds to process with SSDM (as reported in Table 6), is now processed in just 0.03 seconds using the optimized SSDM.
>
> In the optimized version, rather than constructing the label and data kernels $\mathbf{P}$ and $\mathbf{D}$ with dimensions $n \times n$, we randomly sample $k$ labeled samples from the training set. This results in a label kernel $\mathbf{P}$ of size $k \times k$ and a data kernel $\mathbf{D}$ of size $k \times n$, where $k$ is set to $0.01 \cdot n$ if $n > 10,000$ and to $0.1 \cdot n$ if $n \leq 10,000$ in our experiments. Through empirical testing, we found that a small proportion of labeled samples can effectively represent the space in large datasets. If $n$ becomes very large, $k$ can be set to any fixed small value, smaller than $0.01 \cdot n$. Apart from adjusting the kernel dimensions, the other steps of the optimized SSDM remain identical to those in SSDM. This modification reduces the complexity of the optimized SSDM to $O(k^2 \cdot n)$, making it scalable for very large datasets (greater than 100,000 samples). Additionally, the optimized SSDM is implemented using torch, which leverages GPU acceleration.
>
> For comparison, t-SNE has a time complexity of $O(n^2)$, with optimized versions having a complexity of $O(n \cdot \log n)$, while UMAP has a time complexity of $O(d \cdot n^{1.14})$, where $d$ is the dimensionality of the data. As such, for smaller values of $k$, our method is more time-efficient than these alternatives.
>
> In the newly added Appendix F, we provide a comprehensive description of the optimized SSDM, along with detailed performance and runtime comparisons based on the procedure outlined in the Experimental Results section. Additionally, we wish to kindly refer you to our general post in this thread, where we present both evaluation results and runtime comparisons on large datasets, comparing between optimized SSDM, unsupervised Diffusion Maps, and semi-supervised UMAP. In which our SSDM achieves the best performance on 3 out of the 6 datasets, and is significantly faster than semi-supervised UMAP across all datasets. We will upload the code for the optimized SSDM in the supplementary materials within the next few days.
>
> [1] Shen, Chao, and Hau-Tieng Wu. "Scalability and robustness of spectral embedding: landmark diffusion is all you need." Information and Inference: A Journal of the IMA 11.4 (2022): 1527-1595.
>
> [2] Yeh, Sing-Yuan, et al. "Landmark Alternating Diffusion." arXiv preprint arXiv:2404.19649 (2024).

---

> ### Author Response · Authors · 2024-11-20
> **Response to reviewer 7iCi - part 3**
>
> **3. Performance Comparison between SDM and SSDM:** Thank you for the insightful question. The reason we did not directly compare SDM and SSDM is that we considered them suitable for different problem settings. SSDM operates in semi-supervised settings, while SDM in a supervised setting. As such, we considered it more fitting to focus on SSDM in the semi-supervised context and did not directly compare the two methods where SSDM would not be applicable, such as in fully supervised scenarios.
>
> However, we agree that a comparison between SDM and SSDM would be valuable, particularly in semi-supervised settings, and where runtime is not a limiting factor. To address this, we have added such a comparison in Table 5 of the new Appendix E.3. This table includes results for both SDM and SSDM across all datasets for which results were previously reported only for SDM. As shown in the table, SDM slightly outperforms SSDM in most cases, aligning with our theoretical analysis. However, there are cases where SSDM can exceed SDM's performance. For instance, SSDM outperforms SDM on the Yacht dataset and the toy problem presented in the paper, as illustrated in Figure 3.
>
> To better understand why SSDM can sometimes surpass SDM, we examined the components of the toy problem in Figures 11 and 12. In Figure 11, which shows SDM results with the optimal $t = 0.02$, and Figure 12, which shows SSDM results with the optimal $t = 0.8$, we observe that SDM embeddings, which are influenced by individual kernels for each data point, exhibit greater variability. This results in embeddings that are less consistent across samples. In contrast, the embeddings derived using SSDM are more consistent, as they are constructed from a single kernel.
>
> Theoretically, this variability in SDM could be mitigated by using larger kernels. However, as we discussed earlier, SDM becomes impractical for large datasets due to its computational complexity.
>
> In conclusion, while SSDM is significantly more time-efficient and its performance is comparable to SDM, SDM remains relevant for fully supervised settings and can achieve better performance in some cases. We have included a detailed analysis and conclusions regarding this comparison in Appendix E.3 of the revised paper.

---

> ### Comment · Reviewer_7iCi · 2024-12-01
> **Thanks for the detailed response**
>
> Given the response above which has addressed my major concerns, I am happy to increase my score.

---

### Author Response · Authors · 2024-11-20
**Additional experimental results including large scale datasets**

In this post, we present new evaluation results on large datasets, which were not included in the original submission. These results were added to the revised version of our paper in the new Appendix F. By implementing an optimized version of SSDM, as described in Appendix F, we were able to evaluate large datasets effectively.
The new results on large datasets are presented in two tables:

1. **Table 1** shows the performance of our optimized SSDM compared to unsupervised Diffusion Maps and semi-supervised UMAP, following the procedure outlined in the Experimental Results section. We report the Misclassification Rate for each method, and our SSDM achieves the best performance on 3 out of 6 datasets.

2. **Table 2** reports runtime comparisons in seconds between our SSDM and semi-supervised UMAP. The results demonstrate that our SSDM is significantly faster across all datasets, consistent with the complexity analysis in Appendix F.

### Table 1

| Dataset       | n      | d   | DM                         | SSUMAP                     | SSDM (ours)                |
|---------------|--------|-----|:--------------------------:|:--------------------------:|:--------------------------:|
| MNIST         | 70,000 | 784 | 0.22 ± 0.006               | **0.046 ± 0.001**          | 0.1 ± 0.003                |
| Fashion-MNIST | 70,000 | 784 | 0.33 ± 0.016               | **0.21 ± 0.006**           | 0.25 ± 0.004               |
| Isolet        | 7797   | 617 | 0.147 ± 0.003              | 0.178 ± 0.005              | **0.087 ± 0.006**          |
| Adult         | 48,842 | 14  | **0.211 ± 0.002**          | 0.212 ± 0.001              | 0.23 ± 0.003               |
| Landsat       | 6435   | 36  | 0.181 ± 0.013              | **0.127 ± 0.004**          | **0.127 ± 0.007**          |
| Nursery       | 12,960 | 8   | 0.128 ± 0.03               | 0.193 ± 0.005              | **0.082 ± 0.008**          |

### Table 2

|    Dataset    |    n    |  d  | SSUMAP | SSDM (ours)   |
|--------------|-------|---|:------:|:-------------:|
|     MNIST     | 70,000  | 784 |   35   |    **8.1**    |
| Fashion-MNIST | 70,000  | 784 |   35   |   **8.35**    |
|    Isolet     | 7,797   | 617 |    7   |    **1.2**    |
|    Adult      | 48,842  | 14  |   27   |    **1.4**    |
|   Landsat     | 6,435   | 36  |   6.3  |   **0.12**    |
|   Nursery     | 12,960  | 8   |   5.5  |    **0.4**    |

---

### Author Response · Authors · 2024-12-02

We sincerely thank the reviewers for their time and effort in evaluating our work and providing valuable feedback. Their constructive comments have significantly helped us refine and improve our paper.

---

### Meta-Review · Area_Chair_uAt1 · 2024-12-20

**Metareview:**

The reviewers originally raised concerns regarding several aspects such as methodological justification, empirical evaluation and technical novelty. Fortunately, the authors response positively to the reviews, and now all reviewers have reach consensus that the paper should be accepted. In this case, I am happy to recommend accepting the paper.

**Additional Comments On Reviewer Discussion:**

NA

---

### Decision · Program_Chairs · 2025-01-22

Accept (Poster)